# *Aedes aegypti* salivary gland extract enhances Zika virus replication through immune modulation

**Gabriel Hilario[1,8], Gilson Pires Dorneles[2], Deise Nascimento de Freitas[1], Onilda Santos da Silva[3], Josiane Somariva Prophiro[4], Tiago Fazolo[1], Rafael Rahal Guaragna Machado[5], Cristina Bonorino[1,2], Danielle Bruna Leal Oliveira[5,6], Simone Gonçalves da Fonseca[7], Pedro Roosevelt Torres Romão[1,2]/+, Luiz Carlos Rodrigues Jr[1,8]/+**

[1]Universidade Federal de Ciências da Saúde de Porto Alegre, Programa de Pós-Graduação em Biociências, Porto Alegre, RS, Brasil
[2]Universidade Federal de Ciências da Saúde de Porto Alegre, Programa de Pós-Graduação em Ciências da Saúde, Porto Alegre, RS, Brasil
[3]Universidade Federal do Rio Grande do Sul, Programa de Pós-Graduação em Microbiologia Agrícola e do Ambiente, Porto Alegre, RS, Brasil
[4]Universidade do Sul de Santa Catarina, Programa de Pós-Graduação em Ciências da Saúde, Tubarão, SC, Brasil
[5]Universidade de São Paulo, Instituto de Ciências Biomédicas, Laboratório de Virologia Clínica e Molecular, São Paulo, SP, Brasil
[6]Hospital Israelita Albert Einstein, São Paulo, SP, Brasil
[7]Universidade Federal de Goiás, Instituto de Patologia Tropical e Saúde Pública, Goiânia, GO, Brasil
[8]Universidade Federal de Ciências da Saúde de Porto Alegre, Laboratório de Imunovirologia, Porto Alegre, RS, Brasil

**BACKGROUND** Mosquito saliva contains bioactive molecules that modulate host immunity and may influence arboviral infection. The contribution of *Aedes aegypti* salivary gland extract (SGE) to viral replication and immune regulation during Zika virus (ZIKV) infection remains poorly understood.

**OBJECTIVES** To investigate the immunomodulatory effects of *Ae. aegypti* SGE during ZIKV infection.

**METHODS** Peripheral blood mononuclear cells (PBMCs) and murine antigen-presenting cell lines were exposed to ZIKV with or without SGE. Viral replication was measured by quantitative polymerase chain reaction (qPCR), cell death and immune subsets by flow cytometry, oxidative stress markers by biochemical assays, and cytokine production by enzyme-linked immunosorbent assay (ELISA).

**FINDINGS** SGE enhanced ZIKV replication, particularly in PBMCs, with increased RNA copies (median Δ = +1,779), reduced late apoptosis of CD4+ T cells (p = 0.0055), dendritic cell death (p < 0.01), and impaired T-cell proliferation. SGE attenuated ZIKV-induced oxidative damage by restoring glutathione levels, reducing lipid and protein oxidation (p < 0.001), and increasing nitric oxide (NO) production. Cytokine profiling revealed suppression of interferon-γ (IFN-γ) (p < 0.001) and induction of interleukin-4 (IL-4) (p < 0.0001), indicating a Th2-skewed response. Murine cell lines confirmed SGE-driven cytokine modulation.

**MAIN CONCLUSIONS** *Ae. aegypti* SGE alters host immune homeostasis, favouring ZIKV infection by weakening antiviral defences and redirecting immune and redox pathways, thereby facilitating viral expansion.

Key words: Zika virus infection - vector saliva - immunomodulation - arbovirus pathogenesis - leucocytes

The worldwide epidemiological landscape is increasingly burdened by arthropod- borne diseases, among which the Zika virus (ZIKV) has assumed a prominent role. This virus has demonstrated widespread geographical dispersion, affecting various continents including South America, Central America, the Caribbean, and parts of Africa and Asia.[1] ZIKV has been responsible for significant outbreaks in these regions, emerging as a particularly concerning pathogen due to its association with severe neurological complications and congenital anomalies, such as microcephaly in neonates, and Guillain-Barré syndrome in adults, a serious immune response that leads to muscle weakness and paralysis.[1,2] The transmission dynamics of ZIKV are intricately linked to the haematophagic activity of *Aedes aegypti*, a vector that delivers immunomodulatory biomolecules through its saliva into the host's integument; these salivary components are essential in enhancing the viral transmission efficiency.[3,4,5] Unlike earlier studies predominantly focused on Dengue virus (DENV)[6,7,8] and Yellow Fever virus (YFV),[9] the present work specifically investigates the integrated immune and oxidative responses during ZIKV infection in the context of *Ae. aegypti* salivary exposure. Additionally, this study elucidates a previously poorly understood role of *Ae. aegypti* salivary gland extract (SGE) in modulating antigen-presenting cell responses during ZIKV infection, uncovering mechanisms which may inform strategies to control viral replication.

**doi:** 10.1590/0074-02760250272
**Financial support:** FAPERGS, CAPES, INCT PROBIAM.
**+ Corresponding authors:** luizcrj@ufcspa.edu.br | ⓘ https://orcid.org/0000-0002-5380-8930 / pedror@ufcspa.edu.br| ⓘ https://orcid.org/0000-0002-1039-2509

**Handling editor:** Ademir de Jesus Martins Jr | ⓘ https://orcid.org/0000-0001-5739-1215

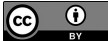

The immunomodulatory properties of mosquito saliva have attracted considerable scientific interest, especially in the context of their influence on viral pathogenesis.[10,11,12] The salivary glands of *Ae. aegypti* harbour an assortment of bioactive constituents capable of modulating host's immune responses, thereby orchestrating a complex interplay between the mosquito vector and the host immune system, including protease inhibitors, nucleotidases, immunomodulatory proteins, and antioxidant agents.[3,5,13,14,15,16] Although individual components have been described to exert specific effects on immune signalling, proliferation and cell death pathways, the net immunomodulatory effect of mosquito saliva likely results from the synergistic or combined action of these molecules.[3,5,13,14,15,16]

In this study, we investigated the overall impact of unfractionated *Ae. aegypti* SGE on human immune cells, aiming to model the complex interactions occurring during natural exposure.[3,5] Mosquito saliva has been linked to enhanced virus transmission, host susceptibility, disease progression, viraemia levels and mortality. [17] These observations collectively indicate that *Ae. Aegypti* saliva play a crucial role in the pathogenesis and severity of various arboviral diseases, including ZIKV, DENV, West Nile virus (WNV), Rift Valley fever virus (RVF), and Semliki Forest virus (SFV).[18-25] These salivary components are known to augment recruitment of target cells for viral infection, thereby increasing the local pool of infected cells. These molecules concurrently attenuate pro-inflammatory signalling pathways by notably inhibiting Nuclear factor kappa B (NF-κB) transcriptional activity, thus establishing a permissive microenvironment to viral replication.[17,24,26] Moreover, certain constituents of mosquito saliva have been identified as inducers of autophagy and inhibitors of T and B lymphocyte proliferation, and they can initiate apoptotic pathways or suppress expression of type I interferons (IFN).[27,28,29] In turn, these immunomodulatory effects target critical elements of the host antiviral response, emphasising the sophisticated role of mosquito saliva in orchestrating host-virus interactions.[27,28,29]

Furthermore, mosquito saliva can recruit dendritic cells (DCs) and reduce the influx of lymphocytes to the injury site caused by the mosquito.[30] While previous studies have described the immunomodulatory potential of *Ae. aegypti* saliva, the specific effects on distinct immune cell subsets particularly DCs, macrophages, and T lymphocytes remain incompletely understood. Most existing research has focused on isolated pathways or individual molecules rather than evaluating the integrated cellular response during viral infection. Moreover, direct comparative analyses of how mosquito SGE influences the viability, activation profile, and effector functions of these cells during ZIKV infection are lacking.

Given this complex interplay between mosquito saliva and the host immune response, our study aims to explore how these interactions affect immune cells that are recruited upon mosquito bite during ZIKV infection. By focusing on macrophages, DCs, and peripheral blood mononuclear cells (PBMCs), we seek to uncover the nuanced mechanisms through which *Ae. aegypti* saliva modulate these cells *in vitro*. We hypothesise that, *Ae. aegypti* SGE modulates the viability and immune function of antigen-presenting cells and lymphocytes. This modulation can promote ZIKV replication and influences the cellular activation, cell death pathways and cytokine profile. To address this, our study evaluates the effects of SGE on macrophages, DCs, and T lymphocytes, with a focus on cellular viability, activation profiles, and viral replication dynamics. To our knowledge, this is the first study to perform a comparative analysis of the immunomodulatory effect of *Ae. aegypti* saliva during ZIKV infection on antigen-presenting cells which are important for viral infection, DCs, and macrophages, as well as PBMCs.

## SUBJECTS AND METHODS

*Zika virus strain* - In the current study, the ZIKV strain, designated as ZIKV/*H. sapiens*/Brazil/PE243/2015 (abbreviated to ZIKV PE243),[31] was utilised for experimental purposes. This specific viral isolate was kindly provided by Dr Marli Tenório, associated with the research facility Ageu Magalhães/CPqAM, a constituent of the Oswaldo Cruz Foundation (FIOCRUZ) in Pernambuco, Brazil. This strain's utilisation is integral to the study's virological investigations.

*Zika virus production* - ZIKV stocks were initially propagated in *Aedes albopictus* C6/36 mosquito cells and subsequently amplified and titrated in Vero E6 cells (ATCC® CRL-1586™), a lineage derived from African green monkey kidney epithelial cells known for their high susceptibility to flaviviruses and deficient IFN responses. C6/36 cells were maintained in RPMI medium supplemented with 10% foetal bovine serum (FBS) and incubated at 28ºC without $CO_2$. After initial virus propagation, supernatants were harvested and used to infect Vero E6 cells cultured in a 75 cm² culture flask until they achieved approximately 75% confluency. After growth medium removal and phosphate buffered saline (PBS) washing, the cells underwent infection with 200 μL of ZIKV inoculum in 10 mL of serum-free Dulbecco's modified eagle medium (DMEM), followed by incubation (37ºC, 5% CO2) for 2 h. Post initial incubation, the infection medium was substituted with DMEM supplemented with 2% FBS, 50 U/mL penicillin, 50 μg/mL streptomycin, and 15 mM HEPES buffer. Cytopathic effects were observed in about 70% of the cell culture five days post-infection. At this phase, the cells were detached and centrifuged at 1500 rpm for 5 min. The supernatant was collected, and subsequently centrifuged at 5000 rpm for 15 min. The supernatant was aliquoted and stored at -80ºC for future experiments. The virus title was determined by plaque assay, and the results were expressed in plate forming units per millilitres [plaque-forming units (PFU)/mL].[31]

*SGE* - Female *Ae. aegypti* mosquitoes from the Rockefeller strain, aged between seven to 10 days, were rendered immobile via exposure to low temperatures. After immobilisation, surface sterilisation was achieved through brief immersion in 70% ethanol. The salivary glands were then meticulously dissected in a PBS solution, with the

harvested glands being stored in aliquots at -80ºC until further processing. The glands underwent a series of three freeze-thaw cycles to prepare the protein extracts, a method employed to disrupt the glandular tissue. The resulting pooled saliva was then subjected to filtration using a 0.22 µm filter. Protein concentration was determined by NanoDrop 2000 (Thermo Fisher Scientific).[32,33,34]

*Effect of Ae. aegypti SGE on ZIKV infection in Vero cells* - First, Vero cells ($1 \times 10^4$/well) were seeded into a 24-well plate until reaching confluence to evaluate the effect of *Ae. aegypti* SGE on the propagation of the ZIKV. ZIKV ($1 \times 10^3$ PFU/mL) and SGE (0.5, 1, 2.5 and 5 µg/mL) were simultaneously added to the cultures. The infection was performed according to the previously described standardised protocol in Vero cells. Cytopathic effects were observed daily, the plaque medium was removed at the end of three days, and the cells were stained with 1% violet crystal to determine the viral titre. The results were expressed in PFU/mL.[31] All experiments included the following control conditions: medium only (mock), SGE alone, ZIKV alone, and ZIKV co-exposed with SGE.

*Peripheral blood mononuclear cells isolation* - PBMCs were isolated from venous blood samples obtained from healthy adult donors (n = 6). All donors were young individuals (aged between 20 and 35 years), with no evidence of active infection, chronic disease, or recent vaccination at the collection time. They exhibited normal body mass index (BMI) values (18.5-24.9 kg/m²) and were not under any immunomodulatory or anti-inflammatory treatments.[31] A total of 10 mL of peripheral blood was collected from the antecubital vein into EDTA-coated tubes. PBMCs were separated by density-gradient centrifugation using Histopaque 1077 (Sigma-Aldrich, St. Louis, MO), following the manufacturer's instructions).[35] After centrifugation, the PBMC layer was carefully collected, washed twice with PBS, and resuspended in complete RPMI-1640 medium (Sigma-Aldrich) supplemented with 10% FBS, 2 mM L-glutamine, 100 U/mL penicillin, and 0.1 mg/mL streptomycin.

Cell viability was assessed using the trypan blue exclusion method, and all preparations consistently presented viability > 95%. Cells were then counted, adjusted to the required concentration for each experiment, and immediately used in subsequent assays.

*Flow cytometric analysis of PBMCs* - PBMCs ($1 \times 10^5$ cells/well) were cultured in RPMI medium 10% FBS, supplemented with interleukin-2 (IL-2) (5 ng/mL) to guarantee lymphocyte survival. Then, the cells were incubated with medium, medium plus SGE 5 µg/mL, or infected with ZIKV [multiplicity of infection (MOI) 0.1] and incubated or not with SGE 5 µg/mL. ZIKV and SGE were added to the cultures at the same time. After 72 h, the cells were prepared for flow cytometric analysis. The antibodies used for the analysis were: cell viability (Amcyan; Cat 564406), HLA-DR (APCH7; Clone G46-6; Cat 561358), Lineage 2 (FITC; Cat 643397), CD141 (PE; Clone 1A4; Cat 559781), CD303 (BV421; Clone V24-785; Cat 566427), and CD11c (PercP-cy5.5; Clone B-Ly6; Cat 565227). Similarly, the following markers were utilised to assess lymphocyte populations: cell viability (Amcyan; Cat 564406), CD3 (PE; Clone Okt3; Cat 317308), CD4 (Percpcy5; Clone rpat4; Cat 560650), CD8 (FITC; Clone H1T8a; Cat 555634), KI67 (APC; Clone b56; Cat 561126), and Granzyme B (BV421; Clone gb11; Cat: 563389).

Samples were immediately acquired on a flow cytometer (BD FACSCanto II™). Data were analysed using FlowJo™ software (version 10, BD Biosciences). Compensation was performed using single-stained controls. Cells were gated to exclude debris, and doublets were eliminated using forward scatter area versus height (FSC-A vs FSC-H) parameters. Frequencies and absolute numbers of each population were calculated and compared across experimental groups. All experiments included the following control conditions: medium only (mock), SGE alone, ZIKV alone, and ZIKV co-exposed with SGE.

*Cell death analysis* - Cells were cultured by 72 h under different experimental conditions (Only media, SGE, ZIKV, and ZIKV+SGE) and cell death assessed by flow cytometry using Annexin V-FITC and propidium iodide (PI) staining. The SGE concentration used in these assays (5 µg/mL) was selected based on our preliminary dose-response experiment. Cells were harvested, washed twice with cold PBS, and resuspended in 1X Annexin V binding buffer (10 mM HEPES, 140 mM NaCl, 2.5 mM CaCl$_2$, pH 7.4). Approximately $1 \times 10^5$ cells per sample were incubated with 5 µL of Annexin V-FITC and 5 µL of PI solution for 15 min at room temperature in the dark, following the manufacturer's instructions (4G Biotech kit). Samples were immediately acquired on a flow cytometer (BD FACSCanto II™) and the data were analysed using FlowJo™ software (version 10, BD Biosciences). Compensation was performed using single-stained controls.

Cells were gated to exclude debris, and doublets were eliminated using forward scatter area versus height (FSC-A vs FSC-H) parameters. Viable cells were identified as Annexin V⁻/PI⁻, early apoptotic cells as Annexin V⁺/PI⁻, late apoptotic/necrotic cells as Annexin V⁺/PI⁺, and necrotic cells as Annexin V⁻/PI⁺. Frequencies and absolute numbers of each population were calculated and compared across experimental groups.

*Quantification of viral RNA by real-time polymerase chain reaction (RT-PCR)* - Viral RNA was extracted from culture supernatants using the MagMAX™ Viral/Pathogen Nucleic Acid Isolation Kit (Applied Biosystems™, Thermo Fisher Scientific™), following the manufacturer's protocol. First, 200 µL of each sample was processed through magnetic bead-based nucleic acid isolation, with subsequent washes and elution in 50 µL of elution buffer. Purified RNA samples were stored at -80ºC until further analysis.

Quantification of ZIKV RNA was performed using the Allplex™ Zika Virus Assay (Seegene®), a multiplex Real-Time PCR kit designed for specific detection of ZIKV RNA. Reactions were prepared according to the manufacturer's instructions, using 5 µL of extracted RNA per reaction. Amplifications were performedin a CFX96™ Real-Time PCR Detection System (Bio-Rad™), with the following cycling conditions: reverse transcription at

50ºC for 20 min, initial denaturation at 95ºC for 15 min, followed by 45 cycles of 95ºC for 15 s and 58ºC for 30 s.

The Allplex™ Zika virus assay includes internal controls to monitor extraction efficiency and PCR inhibition. All samples, positive controls, and negative controls were tested in parallel. Data were analysed using Seegene Viewer™ software, and viral loads were calculated based on cycle threshold (Ct) values, expressed as copies per millilitre of supernatant. This approach follows established methodologies previously described for ZIKV RNA quantification, which employ standard curves from RNA transcripts or viral stocks to achieve absolute quantification.[36,37] All experiments included the following control conditions: medium only (mock), SGE alone, ZIKV alone, and ZIKV co-exposed with SGE.

*Effect of SGE on cytokines production by mouse antigen-presenting cells* - The JAWSII mouse dendritic cell line (ATCC® CRL-11904™) and the mouse macrophage cell line RAW 264.7 mouse macrophage cell line (ATCC® TIB-71™) were used to determine the influence of SGE on cytokine production by antigen-present cells upon ZIKV infection. JAWS II were maintained in alpha minimum essential medium (MEM) with ribonucleosides (Sigma), deoxyribonucleosides, 4 mM L-glutamine, 1 mM sodium pyruvate, and 5 ng/mL murine GM-CSF (Peptrotech), 20% FBS, and incubated at 37ºC in 5% of $CO_2$ atmosphere. RAW 264.7 was maintained in DMEM 10% FBS. Each cell line was plated in 24 well plates (1 x $10^4$/ well) and cultured under the following conditions: medium (control), medium plus SGE (1 µg/mL), infected with ZIKV (MOI 0.1), or infected with ZIKV (MOI 0.1) and treated with different concentrations of SGE (0.5 µg/mL, 1 µg/mL, 2.5 µg/mL, and 5 µg/mL). ZIKV and SGE were added to the cultures at the same time. Then, after 90 min of incubation at 37ºC and 5% $CO_2$ with shaking every 15 min, FBS at the final concentration of 10% was added to each well. The supernatants were separated after 72 h of infection and used to quantify IL-1β, tumour necrosis factor-alpha (TNF-α), IL-12, and IL-10 using commercial enzyme-linked immunosorbent assay (ELISA) following the manufacturer's instructions (Thermo Fisher). All experiments included the following control conditions: medium only (mock), SGE alone, ZIKV alone, and ZIKV co-exposed with SGE.

*Effect of SGE on the response of human peripheral blood mononuclear cells* - PBMCs from healthy donors (1 x $10^6$ cells) were initially plated in 300 µL of RPMI-1640 (Gibco) culture medium without FBS. ZIKV was added to the culture at a MOI of 0.1, which corresponded to a 1:8 dilution of the viral stock in the final culture volume, based on prior titration alone or combined with SGE 1 to 5 µg/mL (dissolved in sterile PBS). ZIKV and SGE were added to the cultures at the same time. The final volume of each well was 900 µL/well. Then, after 90 min of incubation at 37ºC and 5% of $CO_2$, with shaking every 15 min, FBS was added to a final volume of 100 µL/well. The plate was centrifuged (500g, 7 min), after 48 h of incubation, and the supernatant and cells were harvested and frozen at -80ºC. The cytokines IL-4 and IFN-γ in the supernatant were quantified by ELISA

following the manufacturer's recommendations (Thermo Fisher). All experiments included the following control conditions: medium only (mock), SGE alone, ZIKV alone, and ZIKV co-exposed with SGE.

*Cell redox state* - Cells were infected and treated or not with SGE 1 to 5 µg/mL as previously described to determine the cell redox state in PBMC upon ZIKV infection. ZIKV and SGE were added to the cultures at the same time. Nest, they were resuspended in PBS 48 h after incubation and submitted to two freeze-thaw cycles to promote cell lysis. The cell lysate was centrifuged (10,000 g, 15 min), and the supernatant was used to determine the redox state. Thiobarbituric acid reactive substance (TBARS) levels were determined according to the method described by Ohkawa et al.[38] The total glutathione (GSH) content was analysed by the method of recycling the 5,5`-dithiobis - [2- nitrobenzoic] acid, DTNB - GSSG proposed by Griffith.[39] Nitrite levels were a reliable measure of nitric oxide (NO) and evaluated through the Griess reagent reaction with the sample, according to the method described by Miranda et al.[40] Plasma advanced oxidation protein products (AOPP) were determined spectrophotometrically according to the method previously described by Henke et al.[41] The results were calculated using the standard curve as the chloramine equivalent and are shown as µmol/L. All experiments included the following control conditions: medium only (mock), SGE alone, ZIKV alone, and ZIKV co-exposed with SGE.

*Statistical analysis* - Results with cell lines are expressed as mean ± standard error of the mean (SEM) from three replicates in three independent experiments and analysed by one-way analysis of variance (ANOVA), followed by Tukey's test. Statistical tests were performed using GraphPad Prism 6.01 software. The results with human cells were analysed using the statistical program SPSS 22.0 (SPSS Inc., USA). Normality of data distribution was assessed using the Shapiro-Wilk test prior to selecting parametric or non-parametric statistical tests. Values were presented as mean ± standard deviation (SD). A one-way ANOVA followed by a Bonferroni post-test was applied to assess the effect of treatments. A significance level of $p < 0.05$ was adopted for all analyses.

*Ethics* - The Ethics Committee of the Hospital e Maternidade Dona Iris, Goiânia, Goiás, Brazil, approved the protocol utilised in the present study, under the protocol CAAE: 62903216.6.0000.8058. All subjects participating in the study provided written informed consent and signed Informed Consent Forms.

## RESULTS

*Aedes aegypti salivary glands improve ZIKV infection in vitro* - We initially performed a plaque reduction assay utilising the susceptible Vero host cell line to evaluate the potential impact of *Ae. aegypti* SGE on ZIKV replication *in vitro*. Remarkably, a dose-dependent increase in the number of PFU/mL of ZIKV was observed in the infected Vero cells, directly correlating with the amount of SGE administered, as shown in Fig. 1A. A statistically significant difference was observed when 5

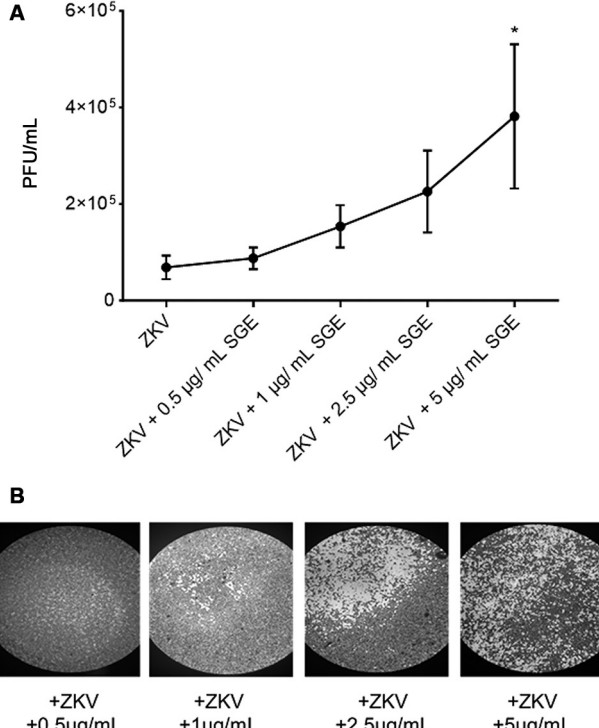

Fig. 1: *Aedes aegypti* salivary gland extract (SGE) enhances Zika virus (ZIKV) replication in Vero cells. (A) Dose-response curve showing plaque-forming units per millilitre (PFU/mL) recovered from Vero cells infected with ZIKV [multiplicity of infection (MOI) = 0.1] in the presence of increasing concentrations of SGE (0.5, 1, 2.5, and 5 µg/mL). Data are expressed as mean ± standard error of the mean (SEM) from three independent experiments (n = 3). Statistical significance was determined by one-way analysis of variance (ANOVA) followed by Tukey's post hoc test. Asterisks indicate statistical significance compared with ZIKV alone (p < 0.05). (B) Representative plaque assay images of Vero cell monolayers infected with ZIKV (MOI 0.1) and treated with SGE at the indicated concentrations, illustrating the dose-dependent increase in cytopathic effects.

mg/mL of SGE was used in the treatment. Importantly, SGE alone did not induce cytopathic effects in Vero cells under any of the tested concentrations (0.5-5 µg/mL).

Figure 1B shows representative images of Vero cells exposed to increasing SGE concentrations during ZIKV infection. This visual evidence demonstrates an enhancement of the cytopathic effects of ZIKV in direct association with increasing amounts of SGE.

*Enhanced late apoptosis and necrosis in PBMCs exposed to ZIKV and Ae. aegypti SGE* - Given that *Ae. aegypti* SGE intensified ZIKV cytopathic effects in Vero cells (Fig. 1), we next examined whether a similar interaction occurs in human PBMC. Leukocyte viability was assessed by flow cytometry 72 h post-infection. SGE alone did not alter cell viability compared with media controls, indicating no intrinsic toxicity. In contrast, ZIKV infection increased absolute numbers of dead leukocytes, and this effect was further enhanced by co-exposure to SGE (Friedman test, p = 0.0129; post-hoc ZIKV+SGE vs SGE, p = 0.0109; Fig. 2A).

Next, we quantified Annexin V+ (early apoptosis), PI+ (late apoptosis/necrosis), and Annexin V+/PI+ (late apoptosis/secondary necrosis) in leukocytes treated with SGE and/or ZIKV to assess cell death modalities. Absolute counts of late apoptotic/necrotic cells were higher in the ZIKV+SGE group than in SGE alone (p = 0.0335), with double- positive late apoptosis/secondary necrosis cells also elevated (post-hoc p = 0.0327) (Fig. 2B). Early apoptosis percentages showed a significant group effect (p < 0.0001) with lower frequencies in ZIKV+SGE compared to SGE and ZIKV alone. In contrast, late apoptotic/necrotic and double-positive cells percentages did not differ significantly among groups (p = 0.187 and p = 0.6685, respectively). These findings underscore the value of using absolute cell counts as the primary readout, while percentage distributions provide complementary insights which mirror the same overall trends. A plausible interpretation is that the ZIKV+SGE condition alters the dynamics of cell death, reducing early apoptosis while increasing late apoptotic/necrotic cells, thereby potentially prolonging the period during which infected cells remain permissive to viral replication. Finally, a representation of cell death dynamics integrating all markers (Fig. 2C) illustrates the overall distribution of early apoptosis, late apoptosis, and necrosis across conditions, confirming that the predominant death profiles under ZIKV infection, with or without SGE, progress toward late apoptotic and necrotic stages. Two-way ANOVA revealed significant effects of condition, marker type, and their interaction (p < 0.0001, p = 0.0216, and p = 0.0257, respectively). Fewer early apoptotic cells coincided with increased late apoptotic/necrotic populations in the ZIKV+SGE group, being consistent with an altered trajectory of leukocyte death under combined exposure.

*Combined ZIKV and SGE exposure alters CD4+ T cell apoptosis without impacting. CD8+ T Cells* - Based on our analysis of total PBMCs, we next assessed cell death markers in CD4+ and CD8+ T cell subsets using samples from five independent donors. Early apoptotic and late apoptotic/necrotic single-positive populations for CD4+ T cells did not differ significantly across conditions (Friedman test, p > 0.05) (Fig. 3A). However, the double-positive population representing late apoptosis/secondary necrosis was significantly affected (p = 0.0055) (Fig. 3A), with post-hoc analysis revealing a reduction in ZIKV+SGE compared with media (p = 0.0197) and SGE (p = 0.0423) (Fig. 3A). Absolute counts mirrored this trend, showing fewer double-positive CD4+ T cells in ZIKV+SGE relative to ZIKV alone (p = 0.0423) (Fig. 3A). In contrast, CD8+ T cells displayed no significant changes in any marker, whether expressed as percentages or absolute numbers (Friedman test, p > 0.05) (Fig. 3B). These findings collectively indicate that combined exposure to ZIKV and *Ae. aegypti* SGE modestly alters CD4+ T cell death profiles, while CD8+ T cells remain largely unaffected (Fig. 3). Importantly, despite the observed decrease in overall PBMC viability under ZIKV exposure, T cells do not appear to be the primary population undergoing cell death. This suggests that other leukocyte subsets, rather than T lymphocytes,

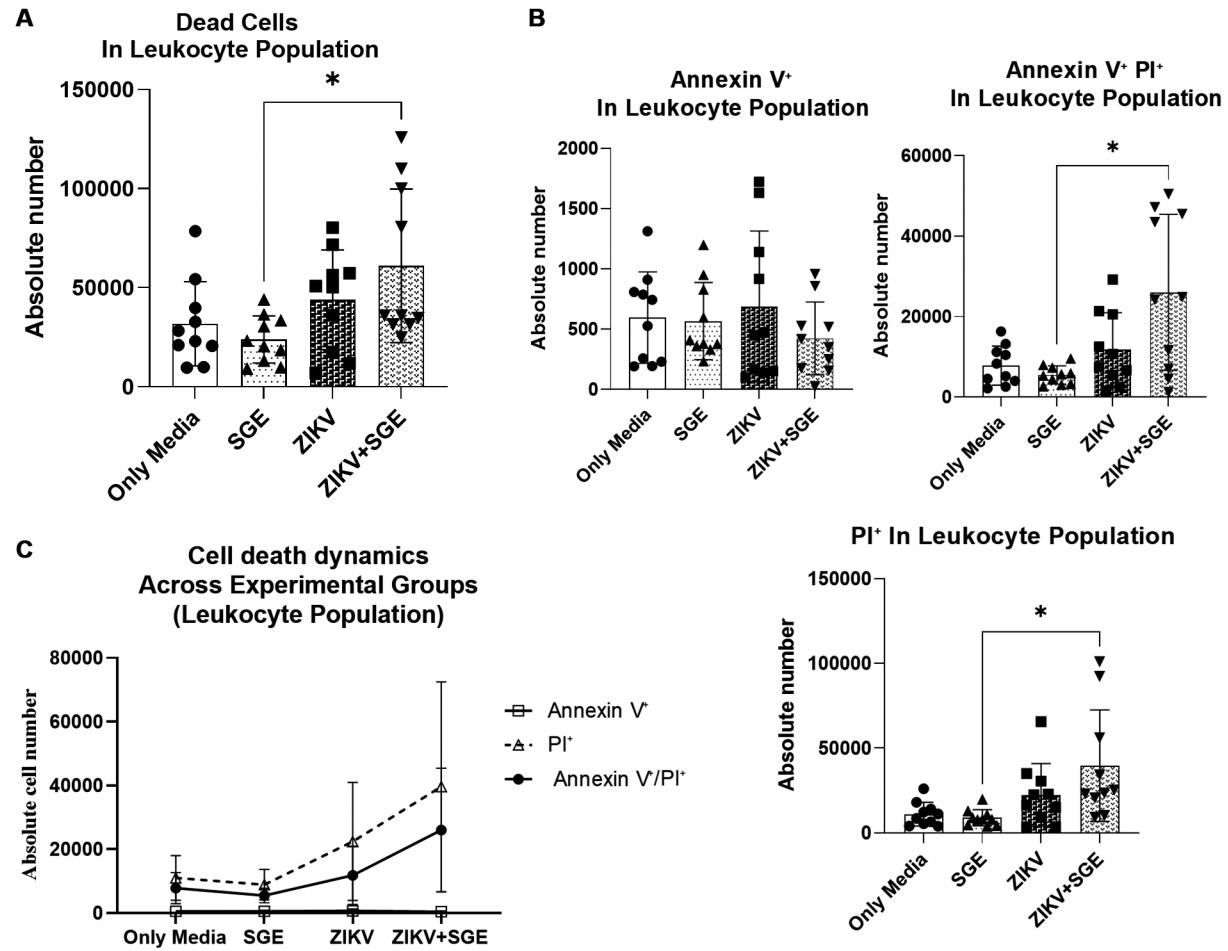

Fig. 2: leukocyte death profiles in peripheral blood mononuclear cells (PBMCs) following Zika virus (ZIKV) infection and *Aedes aegypti* salivary gland extract (SGE) exposure. (A) Absolute number of dead leukocytes after 72 h of culture with medium only, SGE (5 µg/mL), ZIKV, or ZIKV+SGE (5 µg/mL). (B) Absolute counts of Annexin V+ (early apoptotic), PI+ (late apoptotic/necrotic), and Annexin V+/PI+ (late apoptotic/secondary necrotic) leukocytes across groups. (C) Cell death dynamics representation (Annexin V+, PI+, Annexin V+/PI+) illustrating the redistribution of death profiles under different experimental conditions. Data represent PBMCs from five independent donors, each tested in duplicate (individual points shown), with median ± interquartile range (IQR). Statistical comparisons were performed using Friedman test with post-hoc analysis (Panel A: p = 0.0129 overall; ZIKV+SGE vs SGE, p = 0.0109; Panel B: PI+ ZIKV+SGE vs SGE, p = 0.0335; Annexin V+/ PI+ ZIKV+SGE vs SGE, p = 0.0327; Annexin V+ frequencies, p < 0.0001). Two-way analysis of variance (ANOVA) for the Cell death dynamics (Panel C) detected significant main effects of condition and marker type, and their interaction (p < 0.0001, p = 0.0216, p = 0.0257, respectively).

account for the majority of cell loss detected in these assays, highlighting a selective impact of SGE on the immune cell landscape during ZIKV infection.

*ZIKV-induced dendritic cell death is attenuated by Ae. aegypti SGE* - Extending our analysis of leukocyte death under ZIKV and SGE co-exposure we next evaluated death markers in DCs 72 h after incubation with ZIKV, SGE, or both in combination. Absolute counts of early apoptotic DCs differed significantly among groups (Friedman test, p < 0.0001) with reduced numbers in ZIKV+SGE compared to media (p = 0.0014) (Fig. 4). Late apoptotic/necrotic DCs also showed significant group differences (p = 0.0087) (Fig. 4), and post-hoc testing showed that ZIKV alone increased PI+ cells compared to ZIKV+SGE (p = 0.0197) (Fig. 4). Similarly, double- positive late apoptosis/secondary necrosis DCs varied across conditions (p = 0.0023), with higher counts

in ZIKV compared to ZIKV+SGE (p = 0.0087) (Fig. 4). Percentage analyses supported these trends, though the magnitude of differences was more evident in absolute values (Annexin V+, p < 0.0001; PI+, p = 0.0226; Annexin V+/PI+, p = 0.0167). Together, these data show that ZIKV infection increased apoptotic and necrotic markers in DCs, while combined exposure with SGE was associated with reduced accumulation of Annexin V+, PI+, and Annexin V+/PI+ cells.

*Aedes aegypti SGE potentiate ZIKV replication in human PBMCs* - Given the increased cytopathic effects observed in Vero cells and the altered death profiles detected in PBMCs following combined ZIKV and SGE exposure, we next investigated whether these phenomena correlated with changes in viral replication. To do so, ZIKV RNA levels were quantified by qPCR after 72 h of culture under the indicated conditions.

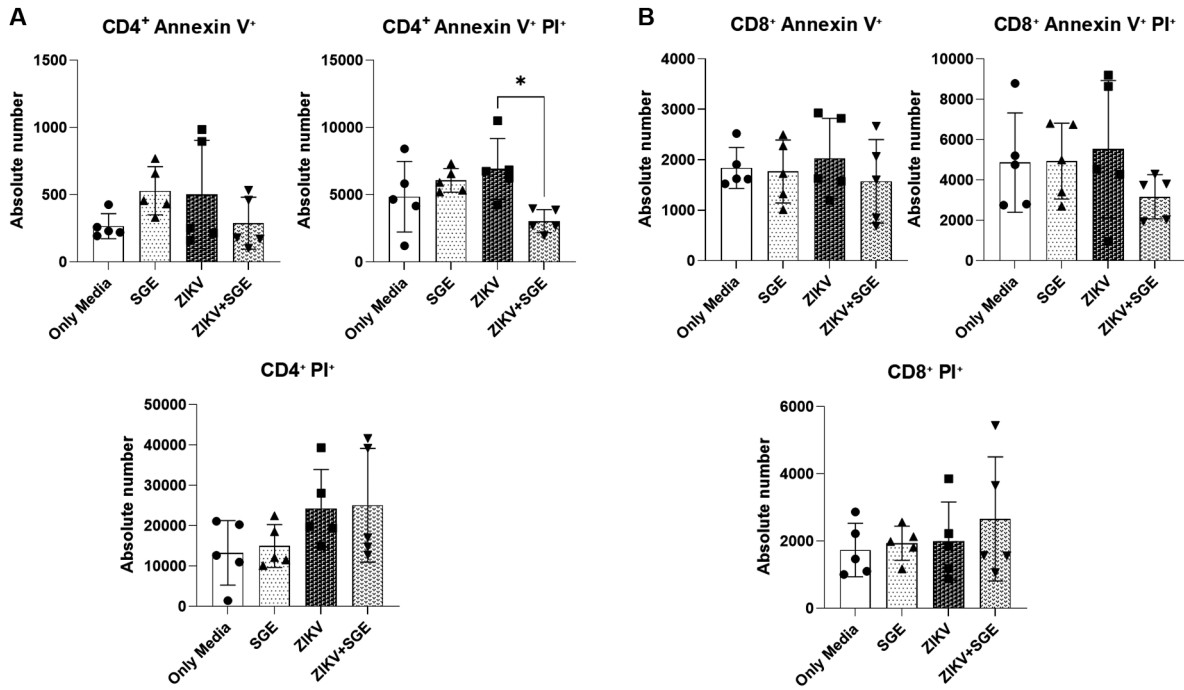

Fig. 3: Zika virus (ZIKV) and salivary gland extract (SGE) co-exposure reduces late apoptotic CD4+ T cells while sparing CD8+ populations. (A) CD4+ T cells: absolute numbers of Annexin V+ (early apoptosis), PI+ (late apoptosis/necrosis), and Annexin V+/PI+ (late apoptosis/secondary necrosis) populations after 72 h under the indicated conditions (Media, SGE (5 µg/mL), ZIKV, ZIKV+SGE (5 µg/mL)). A significant difference was detected for the Annexin V+/PI+ population, with reductions in ZIKV+SGE compared with ZIKV (p = 0.0423). (B) CD8+ T cells: no significant differences were observed across groups for any marker in absolute numbers (Friedman test, p > 0.05). Data represent peripheral blood mononuclear cells (PBMCs) from five independent donors shown as individual values with median ± interquartile range (IQR).

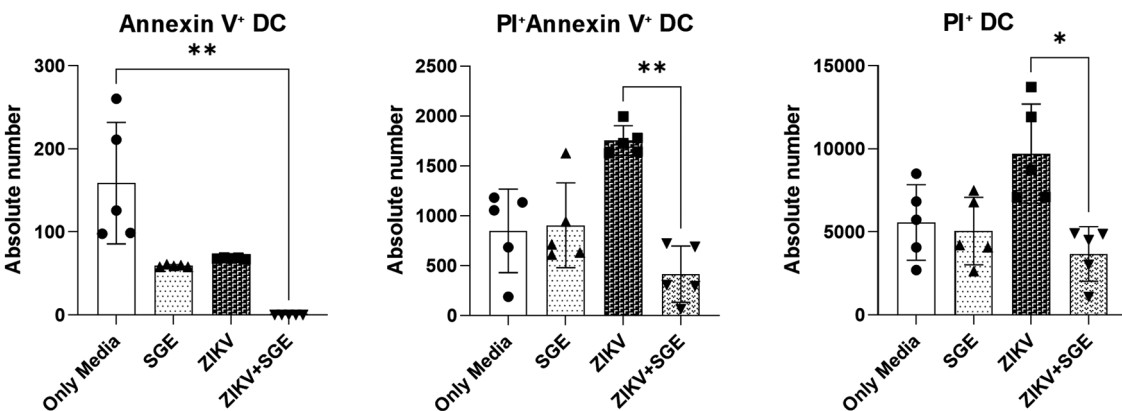

Fig. 4: Zika virus (ZIKV)-induced dendritic cell death is attenuated by *Aedes aegypti* salivary gland extract (SGE). Absolute numbers of dendritic cells positive for Annexin V+ (early apoptosis), PI+ (late apoptosis/necrosis), and Annexin V+/PI+ (late apoptosis/secondary necrosis) were quantified after 72 h of culture under the indicated conditions (Media, SGE (5 µg/mL), ZIKV, ZIKV+SGE (5 µg/mL)). Significant overall differences were observed for early apoptosis (p < 0.0001), late apoptosis/necrosis (p = 0.0087), and late apoptosis/secondary necrosis (p = 0.0023). Post-hoc analyses showed reduced early apoptosis counts in ZIKV+SGE compared to Media (p = 0.0014), fewer late apoptosis/necrosis cells in ZIKV+SGE versus ZIKV (p = 0.0197), and lower late apoptosis/secondary necrosis counts in ZIKV+SGE compared to ZIKV (p = 0.0087). Percentage analyses yielded consistent trends (Annexin V+, *p* < 0.0001; PI+, p = 0.0226; Annexin V+/PI+, p = 0.0167). Data represent peripheral blood mononuclear cells (PBMCs)-derived dendritic cells from five independent donors (no technical replicates), shown as individual values with median ± interquartile range (IQR).

As expected, ZIKV infection led to detectable viral replication in PBMCs, whereas media-only, SGE alone, or inactivated ZIKV controls showed no measurable viral loads (Fig. 5A). A significant main effect was observed across experimental groups (Kruskal-Wallis, p < 0.0001) (Fig. 5A). Post-hoc analyses confirmed significantly higher RNA copy numbers in both ZIKV and ZIKV+SGE groups compared to controls, with ZIKV+SGE exhibiting a markedly greater viral load than ZIKV alone (adjusted p < 0.0001) (Fig. 5A).

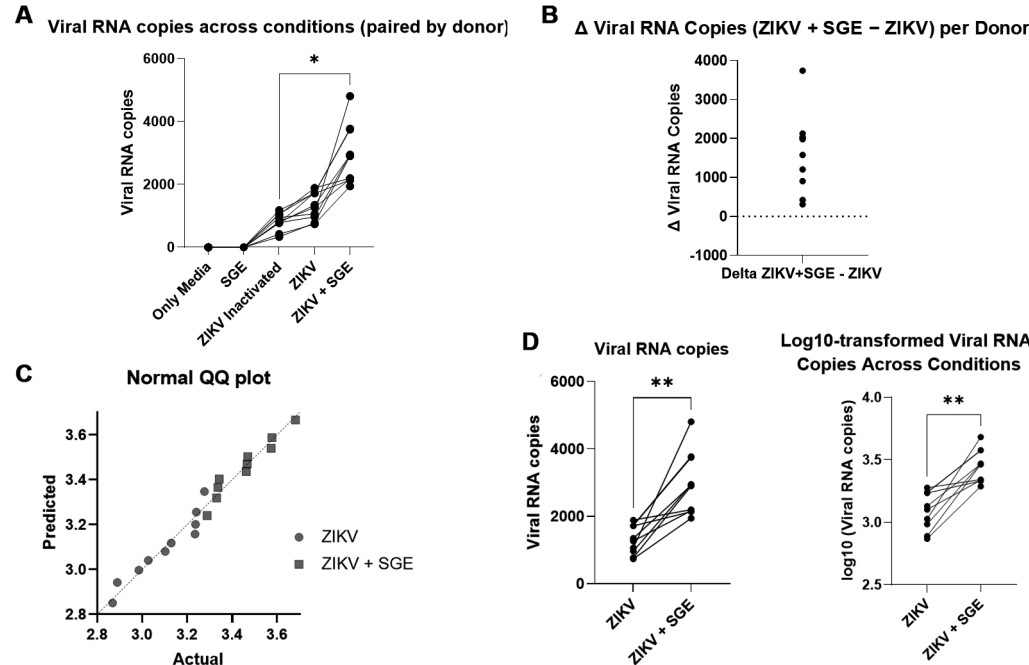

Fig. 5: salivary gland extract (SGE) enhances Zika virus (ZIKV) replication in human peripheral blood mononuclear cells (PBMCs). (A) Absolute viral RNA copy numbers in PBMCs cultured under the indicated conditions (Media, SGE, ZIKV, ZIKV+SGE) for 72 h. Each symbol represents one culture replicate; two replicates were performed per donor, resulting in 10 points per group from five independent donors. Quantitative polymerase chain reaction (qPCR) was performed in technical triplicates and averaged per well. (B) Δ viral RNA copies (ZIKV+SGE - ZIKV) calculated for each donor, showing consistent increases across all individuals. (C) Assessment of normality of viral RNA values by Shapiro-Wilk test (p > 0.05), supporting the use of paired analysis. (D) Paired comparison of viral RNA levels between ZIKV and ZIKV+SGE conditions across donors, shown as both absolute copy numbers and log10-transformed values. Wilcoxon matched-pairs test confirmed significantly higher viral loads in ZIKV+SGE (p = 0.002), with a median increase of 1,779 copies per donor (log10 median difference = 0.3367). Data represent PBMCs from five independent donors; each condition was cultured in duplicate, and qPCR performed in technical triplicates. Points depict culture-level replicates, while statistical analyses were performed on donor-level aggregates (duplicates averaged per donor).

In turn, viral RNA values were further analysed at the individual level to control for inter-donor variability. Thus, Δ viral RNA (ZIKV+SGE - ZIKV) was consistently positive for each of the five donors (in duplicate), indicating a reproducible enhancement across all individuals (Fig. 5B). Normality testing (Shapiro-Wilk, p > 0.05) justified paired analysis (Fig. 5C), and Wilcoxon matched-pairs testing demonstrated a significant increase in the ZIKV+SGE condition compared with ZIKV alone (p = 0.002), with a median increase of 1,779 RNA copies per donor (Fig. 5B). Log-transformation of values yielded the same result (p = 0.002), with a median log10 difference of 0.3367 (Fig. 5D).

These results collectively demonstrate that, *Ae. aegypti* SGE significantly enhances ZIKV replication in human PBMCs. This effect parallels our observations in Vero cells, where SGE promoted ZIKV-induced cytopathic effects in a dose-dependent manner, and supports the conclusion that mosquito salivary components can potentiate viral amplification in primary human leukocytes.

*Aedes aegypti SGE modulates oxidative stress to delay leukocyte apoptosis and promote ZIKV replication* - Given that oxidative stress is a key trigger of apoptosis in immune cells, the modulation of redox homeostasis by *Ae. aegypti* SGE may directly influence leukocyte survival during ZIKV infection. Therefore, were associated with changes in oxidative stress, we measured lipid peroxidation and intracellular redox markers in PBMCs after 72 h of infection *in vitro* to determine whether the combined effects of ZIKV and SGE observed on cell death and viral replication were associated with changes in oxidative stress. ZIKV infection significantly increased TBARS levels compared with controls (p < 0.001), indicating elevated lipid damage. In contrast, co-treatment with SGE markedly reduced thiobarbituric acid reactive substance (TBARS) production (Fig. 6).

Next, we evaluated additional redox markers in a complementary analyses, including intracellular GSH, AOPP, and NO. ZIKV infection decreased GSH levels (p < 0.01) and increased AOPP (p < 0.001), consistent with oxidative imbalance, whereas the presence of SGE restored GSH content and attenuated AOPP accumulation. Notably, NO production was significantly enhanced in ZIKV+SGE compared with ZIKV alone (p < 0.01) (Fig. 7). Together, these results demonstrate that ZIKV disrupts redox homeostasis in PBMCs, and that SGE modulates this response, mitigating oxidative damage. This redox modulation likely delays apoptosis, prolonging leukocyte viability and creating a cellular environment permissive to viral replication, thereby linking the antioxidant effects of mosquito saliva to its pro-viral and immunomodulatory actions.

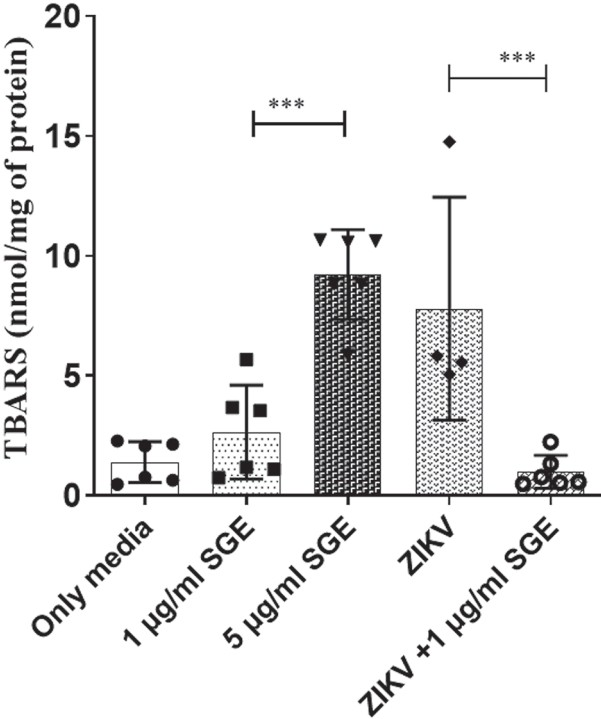

Fig. 6: lipid peroxidation during Zika virus (ZIKV) infection is reduced by *Aedes aegypti* salivary gland extract (SGE). Thiobarbituric acid reactive substance (TBARS) levels (nmol/mg of protein) were quantified in peripheral blood mononuclear cells (PBMCs) cultures after 72 h under the following conditions: control medium, SGE (1 μg/mL), SGE (5 μg/mL), ZIKV, and ZIKV+SGE (1 μg/mL). ZIKV infection significantly increased TBARS relative to controls (**p < 0.001), while co-treatment with SGE (1 μg/mL) significantly reduced TBARS compared with ZIKV alone (**p < 0.001). Data represent PBMCs from six independent donors (individual values shown) with mean ± standard error of the mean (SEM). Statistical analysis: Kruskal-Wallis with Dunn's post-hoc test; asterisks denote significant pairwise comparisons as indicated.

*SGE modulates CD4+ and CD8+ T cell frequencies without reversing ZIKV-induced functional impairment* - Extending the observed effects of ZIKV and SGE on PBMC viability and death profiles, we next investigated whether these conditions altered T cell subsets. As a result, CD4+ T cell frequencies differed significantly across groups (Friedman test, p = 0.0087), after 72 h of culture, driven by an increased proportion of CD4+ cells in the SGE-treated condition relative to the medium control (p = 0.0197) (Fig. 8A). Although absolute CD4+ counts showed a similar upward trend, the difference did not reach statistical significance (p = 0.0548) (Fig. 8B). Proliferative activity was reduced in ZIKV- and ZIKV+SGE-exposed cultures, as indicated by lower frequencies of Ki-67+ CD4+ T cells compared with controls (Friedman test, p = 0.012; post-hoc p = 0.0423 for both) (Fig. 8C), while absolute numbers remained unchanged (p = 0.1616) (Fig. 8D). Granzyme B expression was uniformly low across all conditions, with no significant alterations detected in either frequency (p = 0.6112) (Fig. 8E) or absolute numbers (p = 0.1514) (Fig. 8F). Together,

these data indicate that while SGE modestly increases CD4+ T cell representation, ZIKV exposure predominately dampens CD4+ proliferative responses without inducing detectable changes in cytotoxic potential.

Overall frequencies for CD8+ T cells remained unchanged across conditions (p = 0.2496) (Fig. 9A), although absolute counts were significantly increased in the ZIKV+SGE group compared with controls (p = 0.0443; post-hoc p = 0.0423) (Fig. 9B). Functional profiling revealed a reduction in the frequency of granzyme B+ CD8+ T cells following ZIKV infection relative to controls (p = 0.0443; post-hoc p = 0.0423), with no evidence of restoration in the ZIKV+SGE condition (Fig. 9C). Absolute numbers of granzyme B+ cells did not differ (p = 0.2261) (Fig. 9D). Proliferative activity assessed by Ki-67 expression was diminished under ZIKV exposure. Frequencies of Ki-67+ CD8+ cells were lower in the ZIKV group compared to controls (Friedman test, p = 0.0002; post-hoc p = 0.0087) (Fig. 9E), and absolute counts were significantly reduced in ZIKV+SGE compared with controls (p = 0.0014; post-hoc p = 0.0423) (Fig. 9F). These findings collectively indicate that while SGE increases overall CD8+ T cell numbers in the context of ZIKV infection, it does not counteract ZIKV-associated deficits in cytotoxic function or proliferative capacity.

Taken together, these findings show that although modest numerical increases in CD4+ and CD8+ T cell subsets were detected, particularly in the SGE and ZIKV+SGE conditions, ZIKV exposure consistently impaired T cell proliferative capacity (Ki-67) and cytotoxic potential (granzyme B). These results extend the PBMC-level observations by demonstrating that the viability and death alterations described above are accompanied by functional disruptions within T cell compartments, highlighting a broader immunomodulatory impact of ZIKV, with limited compensatory effects from SGE.

*Expansion of DC populations driven by combined ZIKV and Ae. aegypti SGE exposure* - We analysed DC populations in PBMCs after 72 h of culture to investigate the impact of ZIKV and *Ae. aegypti* salivary SGE on DCs dynamics. Total DCs were significantly increased in both frequency and absolute number in the ZIKV+SGE condition compared with media controls (Friedman test, p = 0.0167 and p = 0.0226, respectively) (Fig. 10A).

Analysis of DC subsets revealed that absolute numbers of conventional type 1 dendritic cells (DC1), myeloid dendritic cells (mDC), and plasmacytoid dendritic cells (pDC) were all elevated in ZIKV+SGE compared to controls (p < 0.05), while relative frequencies remained mostly unchanged. The exception was mDCs, which showed a proportional reduction under ZIKV+SGE exposure (p = 0.0443), likely reflecting redistribution within the expanding DC compartment (Fig. 10B-D). In addition to subset frequencies and absolute counts, we evaluated the mean fluorescence intensity (MFI) of CD11c, CD141 and CD303 as an exploratory readout of dendritic cell phenotypic modulation. Phenotypic analysis indicated subtle changes in surface marker expression. CD303 (pDC) showed reduced MFI across conditions (p = 0.0313), while CD141 (DC1) tended toward

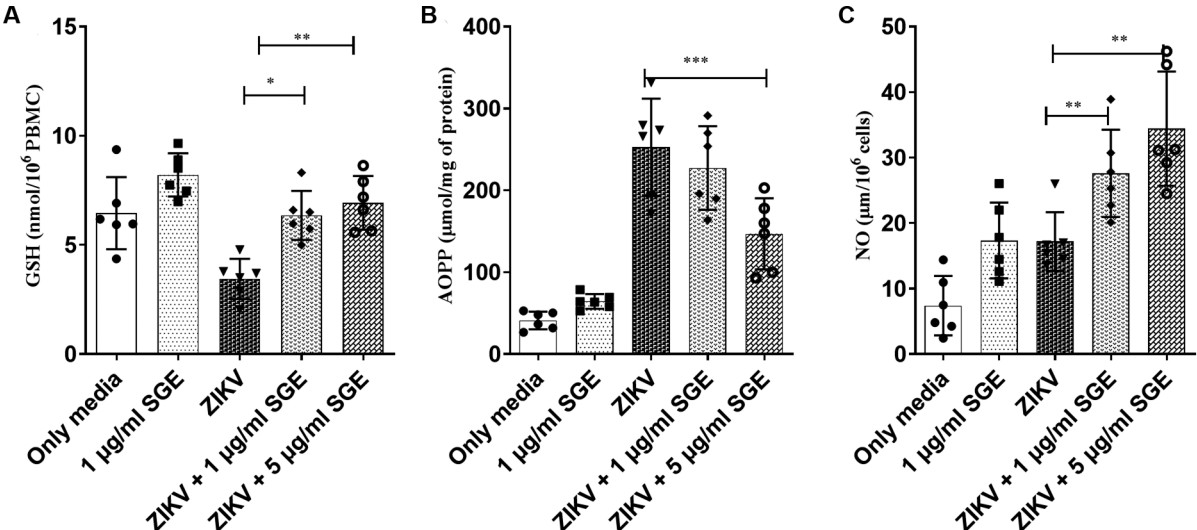

Fig. 7: *Aedes aegypti* salivary gland extract (SGE) modulates oxidative stress markers in Zika virus (ZIKV)-infected peripheral blood mononuclear cells (PBMCs). (A) Total glutathione (GSH) content, (B) advanced oxidised protein products (AOPP), and (C) nitric oxide (NO) levels in PBMC cultures after 72 h under the indicated conditions (Medium, SGE 1 µg/mL, ZIKV, ZIKV+SGE 1 µg/mL, ZIKV+SGE 5 µg/mL). ZIKV infection increased AOPP and decreased GSH compared with controls, consistent with oxidative imbalance. Co-treatment with SGE reduced AOPP levels and restored GSH content. NO production was significantly higher in ZIKV+SGE compared with ZIKV alone (*p < 0.01). Data represent PBMCs from five independent donors (individual values shown) with mean ± standard error of the mean (SEM). Statistical analysis: Kruskal-Wallis with Dunn's post-hoc test; asterisks denote significant pairwise differences.

increased expression in ZIKV and ZIKV+SGE without reaching significance. CD11c (mDC) levels remained unchanged (p = 0.8566) (Fig. 10E). These results indicate that combined exposure to ZIKV and SGE drives absolute expansion of DC populations and subsets, while inducing only minor, non-significant changes in surface marker expression. This suggests that SGE amplifies DC recruitment or survival during ZIKV infection without substantially altering subset phenotypes. (Fig. 10).

*Aedes aegypti SGE promotes a Th2-skewed cytokine profile during ZIKV infection in PBMC cultures* - We analysed the production of IFN-gamma (IFN-γ) and IL- 4 in PBMC cultures from healthy donors after 72 h of stimulation to investigate the influence of *Ae. aegypti* SGE on T helper cytokine polarisation during ZIKV infection.

ZIKV infection alone significantly increased IFN-γ secretion (3.2-fold) compared to media-only controls, reflecting a typical Th1-type response. However, co-exposure to SGE markedly suppressed IFN-γ levels in both infected and uninfected cultures (~2.0- fold, p < 0.001; Fig. 11A), indicating a potent suppressive effect of mosquito salivary components on Th1-type responses. Conversely, SGE significantly enhanced IL-4 production, a hallmark Th2 cytokine, in both ZIKV-infected (2.35-fold) and uninfected controls (4.1-fold) (p < 0.0001; Fig. 11B). demonstrating an active promotion of a Th2- biased environment. Calculation of the IFN-γ: IL-4 ratio further confirmed this shift, with SGE-treated cultures showing a significant reduction in the ratio (p < 0.001; Fig. 11C), reinforcing the shift toward a Th2-dominant immune profile in the presence of mosquito salivary factors during ZIKV infection.

These findings together reveal that *Ae. aegypti* SGE modulates the PBMC cytokine milieu by suppressing Th1-associated responses while enhancing Th2 polarisation, potentially creating an immunological environment that favours viral persistence or dissemination.

*Aedes aegypti SGE modulates cytokine production by antigen-presenting cell lines during ZIKV infection* - Next, we employed the RAW 264.7 macrophage and JAWS II dendritic cell lines as reductionist models of innate immune cells to investigate the direct cellular response to ZIKV and *Ae. aegypti* SGE under controlled conditions. These murine systems are widely used for cytokine profiling and provide high experimental reproducibility, allowing mechanistic interrogation of virus–host interactions. Macrophages were included due to their role as early targets of flavivirus infection and as key mediators of inflammatory signalling. They contribute to both viral replication and the modulation of adaptive immunity, making them an ideal model to assess how SGE components influence innate cytokine responses. In this context, we first analysed the impact of SGE on cytokine secretion by DCs infected with ZIKV at a MOI of 0.1. ZIKV infection alone significantly increased (~1.6 fold) the production of TNF-α, and this effect was further amplified in the presence of SGE (2.1 fold, p < 0.001, Fig. 12A). Similarly, IL-1β secretion was markedly enhanced when SGE was added to ZIKV-infected DCs (16.1-fold, p < 0.0001, Fig. 12B). In contrast, IL-10 production did not show significant modulation following treatment (Fig. 12C). Interestingly, SGE exposure led to a significant reduction in IL-12 levels (~2.78-fold, p < 0.05, Fig. 12D), a cytokine crucial for Th1 differentiation.

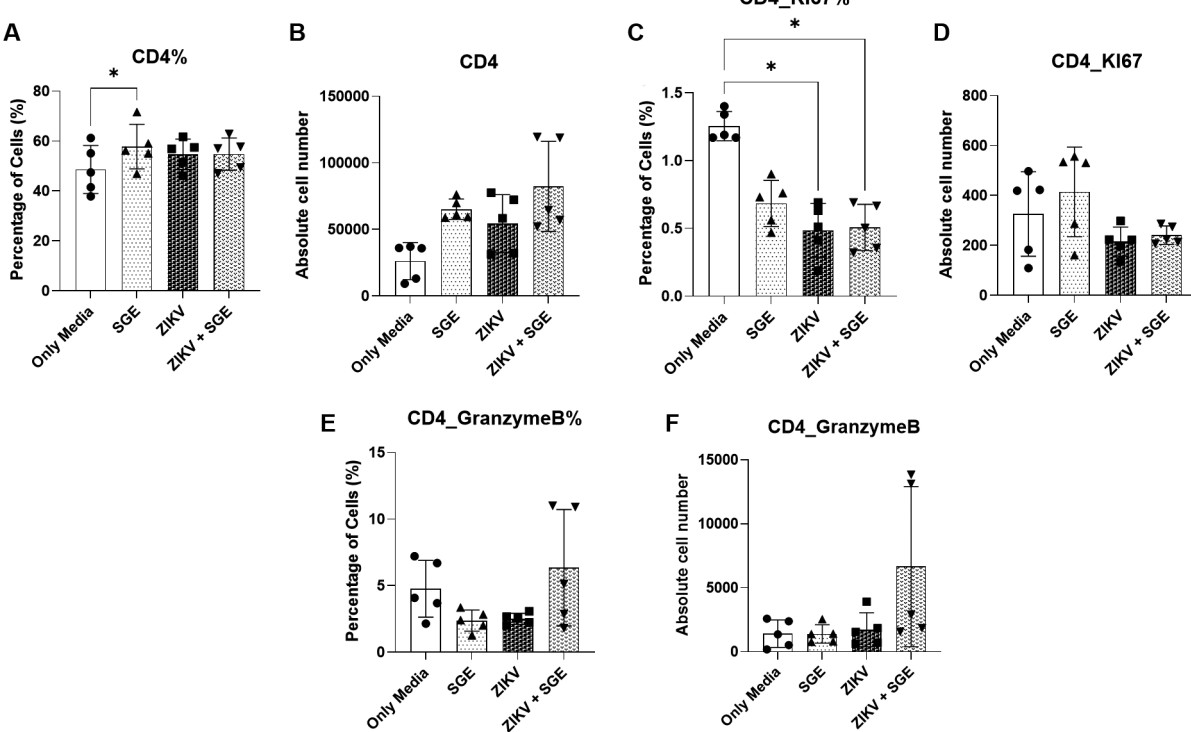

Fig. 8: characterisation of CD4+ T cell frequency, proliferation, and cytotoxic profile following exposure to Zika virus (ZIKV) and/or *Aedes aegypti* salivary gland extract (SGE). (A) Frequency (%) and (B) absolute number of CD4+ T cells in peripheral blood mononuclear cells (PBMCs) cultures after 72 h. (C) Frequency (%) and (D) absolute number of Ki-67+ CD4+ T cells. (E) Frequency (%) and (F) absolute number of granzyme B+ CD4+ T cells. Data were analysed by Friedman test followed by Dunn's multiple comparisons. *p < 0.05; ns: not significant.

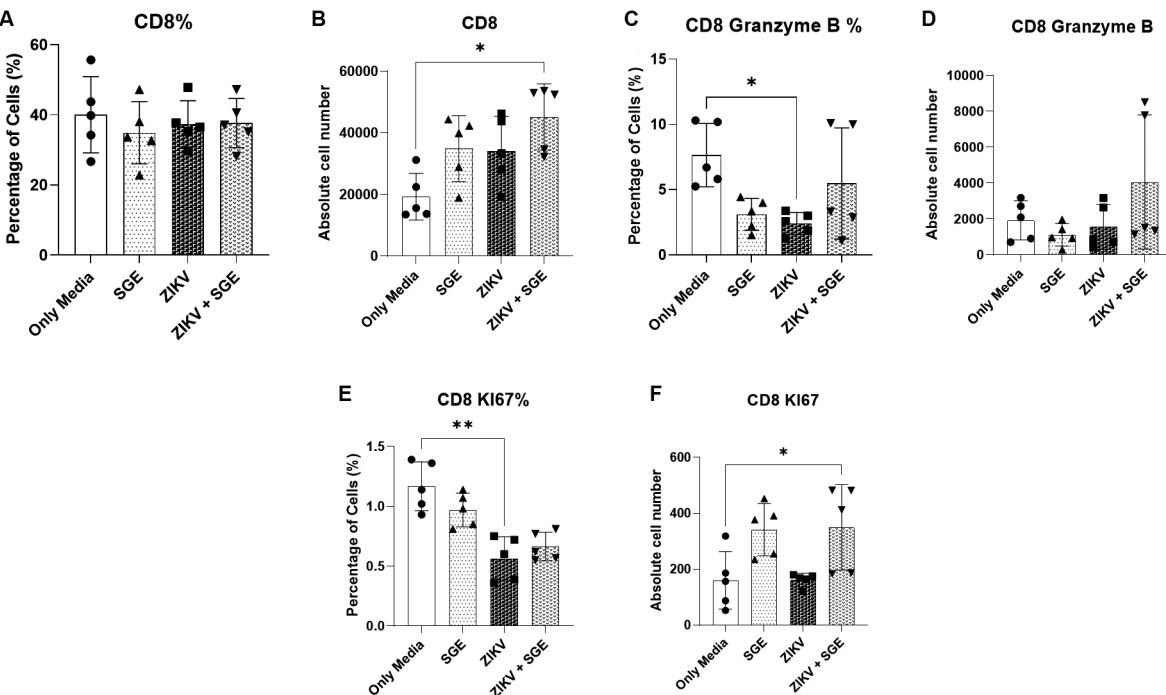

Fig. 9: modulation of CD8+ T cell frequency, absolute number, proliferation, and cytotoxic profile following exposure to Zika virus (ZIKV) and *Aedes aegypti* salivary gland extract (SGE). (A) Frequency of CD8+ T cells among peripheral blood mononuclear cells (PBMCs) after 72 h of treatment. (B) Absolute number of CD8+ T cells in culture. (C) Frequency of granzyme B+ CD8+ T cells. (D) Absolute number of granzyme B+ CD8+ T cells. (E) Frequency of Ki-67+ CD8+ T cells. (F) Absolute number of Ki-67+ CD8+ T cells. Data are presented as individual donors with medians. Statistical analysis was performed using the Friedman test with Dunn's multiple comparisons post hoc test. *p < 0.05.

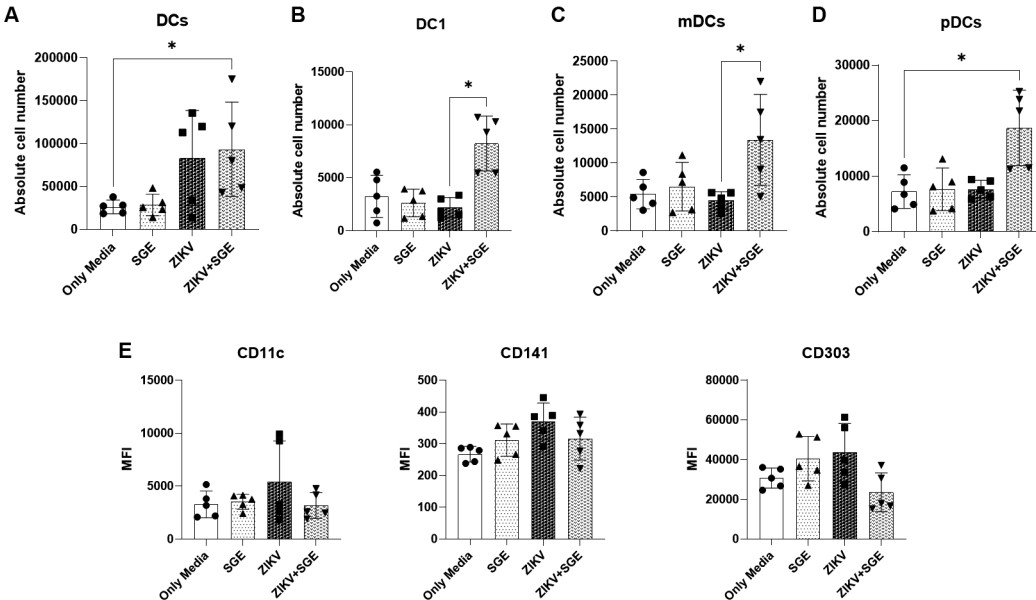

Fig. 10: expansion of dendritic cells and subsets in peripheral blood mononuclear cells (PBMCs) exposed to Zika virus (ZIKV) and *Aedes aegypti* salivary gland extract (SGE). (A) Absolute numbers of total dendritic cells (DCs) and (B-D) subsets (DC1, mDCs, pDCs) after 72 h of culture under the indicated conditions (Only Media, SGE, ZIKV, ZIKV+SGE). Absolute counts of total DCs, DC1, mDCs, and pDCs were significantly increased in the ZIKV+SGE group compared to controls (p < 0.05). (E) Mean fluorescence intensity (MFI) of CD11c (mDCs), CD141 (DC1), and CD303 (pDCs). No significant differences were observed in CD11c or CD141 expression, while CD303 showed a trend toward reduction under ZIKV+SGE exposure. Data represent PBMCs from five independent donors. Individual values are shown with bars indicating median and interquartile range. Statistical analysis: Friedman test with Dunn's post-hoc comparisons; p < 0.05 considered significant.

Parallel experiments conducted in RAW 264.7 macrophages revealed a similar modulatory profile. Co-treatment with SGE and ZIKV resulted in a robust increase in TNF-α (~1.4-fold, p < 0.0001, Fig. 12E) and IL-1β (~3.94-fold, p < 0.05, Fig. 12F) production. Unlike the findings in DCs, IL-12 secretion by ZIKV-infected macrophages was not significantly altered by SGE treatment (Fig. 12G). However, IL-10 production was significantly elevated following SGE exposure at 5 µg/mL compared to lower concentrations and controls (~3.27-fold, p < 0.05, Fig. 12H).

Altogether, these results demonstrate that components of *Ae. aegypti* SGE a potent modulatory effect on key cytokines involved in inflammation and immune regulation during ZIKV infection, promoting a regulatory or Th2-skewed environment. The inclusion of macrophages provides some evidence into how early innate immune responses are shaped by vector-derived factors, complementing observations made in human PBMCs.

## DISCUSSION

ZIKV has developed multiple strategies to evade host immunity, including antagonism of IFN signalling and modulation of cell death pathways, which complicates effective antiviral control.[42-48] Within this context, our study demonstrates that *Ae. aegypti* SGE enhances viral replication and reshapes cellular responses *in vitro*, extending previous observations that mosquito saliva broadly favours arbovirus transmission.[12,25,49,50,51,52]

SGE increased ZIKV plaque formation in Vero cells in a dose-dependent manner, being consistent with prior work showing that mosquito saliva augments infection by flaviviruses such as West Nile and dengue viruses.[12,25] We importantly confirmed this enhancement in primary PBMCs, where co-exposure to ZIKV+SGE yielded higher viral RNA copies across all donors, even after normalisation, highlighting a reproducible effect. As Vero cells are deficient in type I IFN production, the observed enhancement in both systems suggests that SGE can act in early infection steps, possibly by altering cellular susceptibility or the microenvironment, rather than solely via IFN suppression.[49,50,51,52]

Our data indicate that SGE shifted leukocyte death profiles: early apoptotic (Annexin V+) events were reduced, while late apoptotic/necrotic (PI+, Annexin V+/PI+) populations increased in absolute terms. This redistribution suggests that SGE-associated delay in apoptosis may contribute to prolonged cellular viability, which could in turn favour viral replication. Such a pattern is consistent with reports that ZIKV and other flaviviruses actively modulate apoptosis to extend replication time windows.[53] Previous work has shown that *Ae. aegypti* saliva can itself trigger apoptosis in T and B cells,[29] supporting the idea that vector-derived factors influence lymphocyte homeostasis. Our findings suggest that viral and salivary factors may synergistically reshape cell death dynamics under co-exposure to sustain infection.

Reactive oxygen species (ROS) are integral regulators of antiviral immunity and leukocyte homeostasis.[54,55,56,57,58] ROS in PBMCs predominantly derive from phagocyte NADPH oxidases and mitochondrial metabolism, acting both as antimicrobial effectors and as secondary messengers that influence cytokine secre-

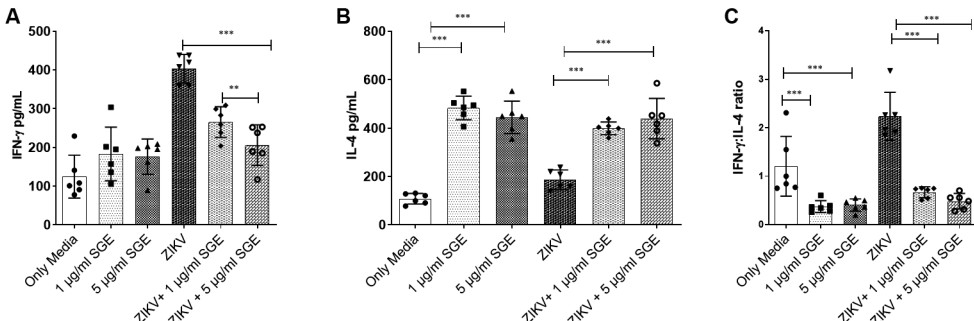

Fig. 11: *Aedes aegypti* salivary gland extract (SGE) skews cytokine responses toward a Th2 profile during Zika virus (ZIKV) infection. Peripheral blood mononuclear cells (PBMCs) from healthy donors were cultured for 72 h with medium alone (Only Media), SGE, ZIKV, or ZIKV+SGE. (A) Interferon-γ (IFN-γ) levels increased after ZIKV infection but were significantly reduced by SGE treatment. (B) Interleukin-4 (IL-4) production was significantly elevated in the presence of SGE, both with and without ZIKV infection. (C) The IFN-γ: IL-4 ratio decreased significantly following SGE treatment, indicating a shift toward a Th2-skewed cytokine profile. Data were analysed using the Friedman test followed by Dunn's multiple comparisons test; *p < 0.05, **p < 0.01, ***p < 0.001.

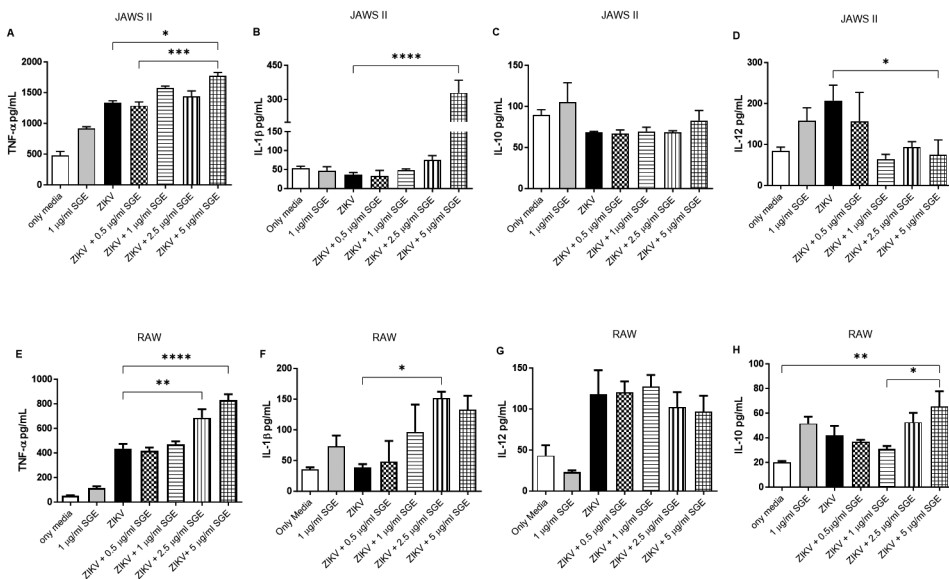

Fig. 12: *Aedes aegypti* salivary gland extract (SGE) modulates cytokine production by antigen-presenting cell lines during Zika virus (ZIKV) infection. JAWS II dendritic cells and RAW 264.7 macrophages were cultured for 72 h under the indicated conditions (Only Media, SGE, ZIKV, or ZIKV+SGE at different concentrations). Cytokine concentrations [tumour necrosis factor-alpha (TNF-α), interleukin-1beta (IL-1β), IL-10, IL-12] were quantified in culture supernatants by enzyme-linked immunosorbent assay (ELISA). In JAWS II cells (A-D), significant differences were observed for TNF-α, IL-1β, and IL-12, while IL-10 levels remained stable. In RAW 264.7 cells (E-H), SGE enhanced TNF-α, IL-1β, and IL-10 secretion, whereas IL-12 showed no significant modulation. Data represent individual values from independent experiments with bars indicating mean ± standard error of the mean (SEM). Statistical analysis was performed using Kruskal-Wallis with Dunn's post-hoc test; p < 0.05 was considered significant.

tion, antigen presentation, and lymphocyte activation.[54,55,56,57,58] Balanced redox signalling is therefore essential for maintaining immune functionality, while excessive oxidative stress can accelerate apoptosis or impair antiviral responses.[54,55,56,57,58] Flaviviruses, including ZIKV, are known to perturb host redox pathways during infection, and mosquito salivary components can modulate intracellular stress responses in innate immune cells.[4,55,56] Within this framework, assessing oxidative stress in mixed PBMC cultures provides a broader view of how viral and salivary factors jointly shape the immune microenvironment during co-exposure.

ZIKV infection alone promoted oxidative imbalance, with elevated lipid and protein oxidation (TBARS, AOPP) and depletion of GSH. Co-exposure to SGE reduced TBARS and AOPP, restored GSH, and significantly increased NO production. Mosquito saliva has been reported to modulate oxidative stress in different contexts, either enhancing or dampening ROS depending on the host stat.[4,59,60] This redox modulation coincided with altered death profiles in our study, suggesting that attenuation of oxidative damage may be associated with delayed apoptosis observed under ZIKV+SGE. While NO possesses antiviral properties, excessive levels can also exert immunoregula-

tory effects,[60] raising the possibility that SGE-associated redox modulation may preserve cell viability and indirectly associated with higher viral replication.

It is also important to consider that the increased viral RNA detected in ZIKV+SGE cultures may partially reflect the preservation of susceptible leukocytes, rather than an isolated increase in intrinsic replication per cell. All PBMC conditions in our experimental design were initiated with identical cell numbers; however, SGE attenuated apoptotic and necrotic death in specific subsets, particularly DCs and CD4+ T cells, which could extend the time window during which these cells remain permissive to infection. Therefore, the enhanced of viral yield observed under SGE exposure is best interpreted as the result of a combined effect involving target-cell preservation and plausible intracellular modulation, without implying a specific dependency mechanism.[53,61,62,63,64]

Numerical changes in T cell subsets were modest in PBMCs: CD4+ frequencies tended to increase with SGE, while CD8+ absolute counts rose under ZIKV+SGE. Because proliferation markers (Ki-67) were reduced, these differences are more consistent with relative preservation of viable subsets rather than true proliferative expansion. Cytotoxic profiles (granzyme B) were functionally reduced in both CD4+ and CD8+ cells during infection. This aligns with previous reports describing that mosquito saliva proteins, including SAAG-4, can skew T helper polarisation and dampen T cell activation. [65,66] Our findings extend this concept by showing that salivary factors not only modulate survival dynamics during ZIKV infection, but also may contribute to a less effective cytotoxic environment, thereby limiting adaptive effector responses.

Dendritic cells are critical for antiviral immunity and are known targets of ZIKV.[67,68,69,70] Total DCs and all subsets (DC1, mDC, pDC) in our assays displayed higher absolute counts under ZIKV+SGE compared with controls. Given that PBMC input numbers were fixed at baseline, and proliferation was not directly assessed, these increases likely reflect relative preservation of DCs rather than *de novo* expansion. In addition to subset frequencies and absolute numbers, we quantified MFI of CD11c, CD141, and CD303 as an exploratory descriptor of DC phenotype, since modulation of surface marker density is frequently used to assess changes in activation or maturation states in human DC subsets. The observed MFI variations were subtle and did not reach statistical significance, indicating that ZIKV+SGE exposure preserved DC subsets without inducing marked phenotypic shifts. Together, these findings support a model in which SGE contributes to DC preservation in ZIKV-infected PBMC cultures, maintaining DC subset composition while exerting only minor effects on their surface phenotype.[67]

ZIKV infection alone elicited a Th1-type cytokine profile in human PBMC cultures characterised by elevated IFN-γ. However, co-exposure to *Ae. aegypti* SGE markedly reduced IFN-γ and increased IL-4, resulting in a decreased IFN-γ: IL-4 ratio. This shift is consistent with a Th2-biased environment, aligning with previous reports that mosquito salivary proteins can skew T help-er polarisation toward IL-4-driven responses.[4,71] Such modulation in primary human leukocytes highlights a central mechanism by which saliva may dampen effective antiviral immunity and favour conditions which support viral persistence.

We evaluated classical antigen-presenting cell lines using JAWS II and RAW 264.7 models as reductionist systems to explore innate immune modulation under controlled conditions to complement the PBMC findings. These models were selected to provide mechanistic insight rather than to serve as translational substitutes for human leukocyte subsets. Macrophages are among the earliest cellular targets of flaviviruses and play central roles in viral replication, inflammasome activation, and pro-inflammatory cytokine production during acute infection.[72,73,74] Therefore, the RAW 264.7 model was included as a complementary approach to assess whether ZIKV-SGE interactions could similarly modulate innate immune activation in a highly permissive myeloid cell type. As a result, SGE co-exposure in JAWS II dendritic cells increased TNF-α and IL-1β while reducing IL-12, whereas SGE enhanced TNF-α, IL-1β, and IL-10 production in RAW macrophages. This pattern suggests an initial pro-inflammatory burst followed by a regulatory shift which can limit Th1 differentiation and CD8+ effector activity. Although derived from murine lines, these results are concordant with human PBMC data and with previous observations in vector-host systems, including sand fly saliva. [34,75,76,77] Together, the PBMC cytokine profile (reduced IFN-γ: IL-4 ratio) and complementary APC-line assays (IL-12↓, IL-10↑) reinforce the interpretation that mosquito saliva drives an immune environment characterised by impaired Th1/antiviral responses and a skew toward regulatory or Th2-like polarisation.

Our data collectively support a model in which *Ae. aegypti* SGE shapes the early host environment by attenuating oxidative stress, delaying apoptotic death, preserving DCs, and limiting T cell proliferation. These convergent effects sustain cellular targets while constraining antiviral effector responses, thereby enhancing viral replication. By integrating these mechanisms, mosquito saliva creates a permissive and regulated environment for arboviral persistence, in line with prior evidence across vector-borne infections.[4,12,16,25,49-52,59,65,66,71,75-79]

This study has some limitations that should be acknowledged. The number of human donors was limited, although the observed effects were consistent across individuals. In addition, the experimental design relied on *in vitro* exposure to SGE, which cannot fully reproduce the complexity of *in vivo* mosquito feeding. The SGE concentrations used reflect commonly employed experimental ranges but may not precisely mirror physiological exposure. Finally, the active salivary components responsible for the observed effects were not identified and warrant further investigation in future studies.

## ACKNOWLEDGEMENTS

To Dra Marli Tenório from Fundação Oswaldo Cuz (FIO-CRUZ) for kindly providing the Zika virus strain.

## AUTHORS' CONTRIBUTION

GM, AH, GD, TF and DF performed experiments and analysed the data; LR and GM designed the work, and wrote and reviewed the manuscript; LR and PR obtaining and managing resources for work; OS and JS maintenance of mosquito colonies and obtaining salivary glands; GM, TF and DF - flow cytometry experiments; DO and RRGM - PCR experiments; PR, DO and OS - manuscript review. All the authors read and approved the final version of the manuscript. All the people involved in the study gave their consent for its publication. The authors declare that they have no competing interests.

## DATA AVAILABILITY

The data supporting the findings of this study are available from the corresponding author upon reasonable request.

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

# OPEN PEER REVIEW

Memórias do IOC thanks the anonymous reviewers for their contribution to the peer review of this work.

**FIRST REVIEW ROUND**

REVIEWERS' COMMENTS

## REVIEWER #1

The authors conducted an evaluation of the effects of Aedes aegypti salivary glands extracts (SGE) on Zika virus infection. Overall the manuscript is well written and the assays support most of the conclusions. Below there are a few comments that, hopefully, will be useful for the authors.

General comments
The authors found that SGE presence resulted in delayed peripheral blood mononuclear cells (PBMCs) apoptosis. They also have found that SGE enhances Zikv replication in PBMCs (Line 384) and stated that SGE modulation of cell machinery for replication (Lines 405 and 544) may be a possible explanation of higher viral load. However the higher viral load could not be a result of higher PBMC cell survivability (and, therefore, greater cell count) rather than cell machinery modulation? If that is the case this possibility should be addressed as well in the replication assay.

Specific comments
Line 156 - Plaque
Line 170 - This was a bit confusing. I believe the viral infection and SGE treatment were simultaneous in this line and in other assays as well. But the way as it is written may also imply a prior treatment with SGE with subsequent viral infection. I would recommend a clarification on this line and in others as well
Line 243 - There are two final points
Line 311 - Plaque
Line 308 - Do SGE alone result in cytopathic effect? If so it would be good to have an image for comparison in figure 1
Figure 7 - There is a problem. The graphs ordinates do not align with the legend. I believe the order of the graphs has been switched.

## REVIEWER #2

This manuscript explores the effects of salivary gland extract (SGE) combined with Zika virus infection on viral replication and immune cell modulation. The authors employ several techniques to compare the combined treatment with infection alone. The manuscript is overall well-written, though some aspects could be modified to improve readability (see minor review suggestions below).

Journal Policy Criteria:
a) Adequacy of the abstract: Well-structured and clear.
b) Originality and importance: This is the first manuscript to explore the combination of SGE with Zika virus in the context of T cell, dendritic cell, and macrophage modulation.
c) Methodology/Results/Discussion: The methodology is well-chosen and appropriately applied. The results are well-written, though some modifications would enhance clarity (see minor review suggestions below). The discussion is very well-developed.
d) References: Meet journal requirements.
e) Figures and tables: Some modifications are needed (see minor review suggestions below).

Minor Review Suggestions and Comments:
Methods (Line 180): The text states that PBMCs were isolated "as described above," but the description actually appears below this reference. Please create a separate methods subsection for PBMC separation and place it as the first method described.
Figure 2C: The term "Cell death cascade" is not ideal, as it suggests the signaling cascade leading to cell death rather than the progression of cell death itself. Consider changing to "Cell death dynamics". Additionally, this result is described in the text as merely repeating findings from panels A and B. However, it actually demonstrates that the overall cell death profiles, due to the infection and the SGE incubation, correspond to late apoptosis and necrosis, a finding that deserves more emphasis.

Line 363: Consider changing "PBMCs from five donors" to "PBMCs from healthy controls" (or simply "PBMCs"). The phrase "five donors" becomes repetitive throughout the text and can be omitted since donor numbers should be specified in the methodology section.

Figure 3:

Clarify what "Abs" means in the figure, preferably in the figure legend.

The CD8 graph title format differs from the CD4 graph; please make them consistent.

The figure legend mentions "frequency," but only absolute numbers are shown.

At the end of this results section, consider adding a concluding statement noting that although overall PBMC death is observed, T cells are not the primary population undergoing cell death, suggesting that other cell types may be dying.

Figure 4:

The description of Annexin+/− and PI+/− combinations is repetitive throughout the text. Consider defining these markers in detail in the first cell death result and subsequently referring to them simply as "early apoptosis," "late apoptosis," or "necrosis."

The SGE concentration is not always clear in some results. Please add this information to the figure legend or clarify in the text that the concentration was determined based on Figure 1 results.

Oxidative stress section: While oxidative stress makes sense within the broader manuscript narrative, this section feels somewhat disconnected. Consider expanding the discussion of how oxidative stress might affect different immune cell populations or influence the biology of the virus/SGE interaction.

DC subset analysis: It is unclear why MFI of CD11c, CD141, and CD303 was measured when these markers were already used to define the populations. I understood that analyzing CD30,3 which is a marker of functionality and type I interferon production on pDC, could indicate something; however, when exploring cytokines, none of the type I interferons were analyzed.

Macrophage data: Although macrophages are important APCs in viral infection, the inclusion of RAW cell data feels somewhat disconnected from the paper, particularly because macrophages were not analyzed in the PBMC experiments. Consider better integrating this data or providing a clearer justification.

Results organization: The discussion is very well-structured, better than the organization of the results section. Consider restructuring the results as follows: Figure 5 should potentially become Figure 2, as it addresses viral replication (a primary outcome). This could be followed by sections exploring cell death and oxidative stress, with an introduction noting that although cells are dying, these are not primarily T cells or DCs. You could then characterize the modifications occurring after the combination treatment. The cytokine data would conclude the results, as it addresses the third signal and T cell polarization.

Summary of my comments:

Comments 1–7 are necessary revisions.

Comment 8 (results reorganization) is a suggestion that would significantly improve the manuscript's flow but is not essential.

## REVIEWER #3

The study entitled "Aedes aegypti salivary gland extract enhances Zika virus replication through immune modulation" addresses an important and original question in arbovirology and vector–host immunology. The work convincingly shows that Aedes aegypti saliva promotes ZIKV replication by altering redox homeostasis, apoptosis, and cytokine polarization in host cells. The experiments are methodologically sound, the results are consistent, and the conclusions are well supported. Nevertheless, several sections require minor textual, structural, and formal adjustments before acceptance:

Abstract:

Page 3, lines 55–58:

The phrase "Aedes aegypti saliva creates a permissive immune environment that favors ZIKV replication by altering cell death dynamics, redox balance, and cytokine polarization" is clear but overly broad. Suggest specifying quantitative information, e.g., "increased ZIKV RNA levels by X "FOLD"... and reduced early apoptosis by "Y%"".

Page 3, line 57:

Consider rephrasing "creates a permissive immune environment" to "promotes conditions conducive to viral replication" to avoid overstatement.

Page 3, line 59:

Keywords are relevant, but "Cytokines" may be redundant since "Immunomodulation" already covers that concept.

Page 4, lines 68–78:

The introduction successfully contextualizes ZIKV transmission but should more explicitly position this study relative to previous work on dengue virus and mosquito saliva. Adding one sentence such as "Unlike earlier studies focused on DENV or WNV, this work explores the integrated immune and oxidative response during ZIKV infection" would clarify novelty.

Methodology, results, and discussion

Page 10–11 (Methods section):

Add information on control conditions, such as mock-infected and heat-inactivated SGE treatments, if used. This ensures reproducibility.

Page 12, lines 245–250:

Statistical analyses should specify whether normality was tested before choosing ANOVA or non-parametric tests. Add a short clarification: "Normality was assessed using Shapiro–Wilk test prior to statistical analysis."

Page 34–35, Figure 1 caption:

Very good caption overall, but include n values for replicates and specify "mean ± SEM from three independent experiments" (as already partly stated). To improve transparency, indicate which test determined significance.

Page 22–24 (Results):

When describing cytokine data, provide numerical fold changes (e.g., "IL-4 increased 2.3-fold compared to control") rather than only qualitative terms ("increased," "decreased").

Page 28–30 (Discussion):

Strengthen interpretation by distinguishing correlation from causation. Example edit at line ~540: change "SGE delays apoptosis, thus facilitating viral replication" to "SGE-associated delay in apoptosis may contribute to prolonged cellular viability, which could in turn favor viral replication."

Page 31–32 (Discussion ending):

Add a short limitations paragraph, e.g., limited donor number, use of in vitro SGE concentrations, and lack of identification of specific active salivary components. (This is optional ok?)

Regarding the figures I have a minor suggestion:

Page 35, Figure 1:

Caption is clear, but add definitions for abbreviations (SGE, MOI, PFU).

Finally, the manuscript is clearly written, but it contains several grammatical inconsistencies, long and complex sentences, and occasional issues with verb tense and article usage. I would strongly recommend a minor english review if it's possible!!

For now, the manuscript is scientifically sound, original, and relevant. Addressing the small textual and methodological clarifications listed above will ensure clarity and compliance with Memórias do Instituto Oswaldo Cruz editorial standards. Congrats to the authors!

## AUTHORS' RESPONSE TO THE REVIEWERS

Dear Editor

Thank you for handling the revision of the manuscript "Aedes aegypti salivary gland extract enhances Zika virus replication through immune modulation." In response to the reviewers' comments, we have revised the manuscript accordingly. We believe that all suggestions have been fully addressed. Below, we provide a detailed, point-by-point response to each reviewer. All line numbers mentioned refer to the revised version of the manuscript.

Reviewer: 1

The authors conducted an evaluation of the effects of Aedes aegypti salivary glands extracts (SGE) on Zika virus infection. Overall the manuscript is well written and the assays support most of the conclusions. Below there are a few comments that, hopefully, will be useful for the authors.

Dear Reviewer, #1,

We thank you for the time devoted to evaluating our manuscript. Your comments have helped improve the quality of our work. Below, we provide our responses to each of your observations.

General comments

The authors found that SGE presence resulted in delayed peripheral blood mononuclear cells (PBMCs) apoptosis. They also have found that SGE enhances Zikv replication in PBMCs (Line 384) and stated that SGE modulation of cell machinery for replication (Lines 405 and 544) may be a possible explanation of higher viral load. However the higher viral load could not be a result of higher PBMC cell survivability (and, therefore, greater cell count) rather than cell machinery modulation? If that is the case this possibility should be addressed as well in the replication assay.

Response: We thank the reviewer for raising this important point. We clarify that all PBMC cultures were seeded with the same initial number of cells ($1 \times 10^6$ cells/well), ensuring that baseline cell numbers were rigorously controlled across all experimental conditions. We agree that the observed increase in viral load may partially result from enhanced PBMC survival. The increased survival of PBMCs and dendritic cells in the presence of SGE, as supported by our Annexin V/PI data, could indeed contribute to a higher overall viral output by preserving a larger pool of susceptible target cells. It is also possible that both mechanisms operate simultaneously.

Importantly, our findings are more consistent with a combined mechanism in which SGE not only preserves cell viability but also modulates antiviral pathways, rather than solely increasing intrinsic viral replication on a per-cell basis. We have addressed this possibility in the Discussion section (line 653 of the revised version).

"It is also important to consider that the increased viral RNA detected in ZIKV+SGE cultures may partially reflect the preservation of susceptible leukocytes rather than an isolated increase in intrinsic replication on a per-cell basis. In our experimental design, all PBMC conditions were initiated with identical cell numbers; however, SGE attenuated apoptotic and necrotic death in specific subsets, particularly dendritic cells and CD4+ T cells, which could extend the time window during which these cells remain permissive to infection. Therefore, the higher viral yields observed in the presence of SGE are likely the result of a combined effect of target-cell preservation and active modulation of antiviral pathways, as suggested by previous reports showing that mosquito salivary components alter host cell death, inflammasome activation, and interferon responses during flavivirus infection"(Page 27, lines 653 to 662).

Specific Comments
Line 156 – "Plaque"
Response: The word "plaque" has been corrected.

Line 170 – This was a bit confusing. I believe the viral infection and SGE treatment were simultaneous in this line and in other assays as well. But the way as it is written may also imply a prior treatment with SGE with subsequent viral infection. I would recommend a clarification on this line and in others as well
Response: In all assays, ZIKV and SGE were added simultaneously, and cells were not pretreated with SGE. This point has been clarified in the Methods section. The revised text (line 189) now reads:
"ZIKV ($1{\times}10^3$ PFU/mL) and SGE (0.5–5 µg/mL) were added simultaneously to the cultures."
To avoid any ambiguity throughout the manuscript, we have also added the clarification "ZIKV and SGE were added to the cultures at the same time" at lines 292, 306 and 318.

Line 243 – There are two final points
Response: The sentence was edited for clarity, as suggested.

Line 311 – "Plaque"
Response: The word "plaque" has been corrected.

Line 308 – Do SGE alone result in cytopathic effect? If so it would be good to have an image for comparison in figure 1
Response: In our assays, SGE alone did not induce visible cytopathic effects in Vero cells at any of the tested concentrations (0.5–5 µg/mL). This information has now been clarified in the Results (line 358):
"Importantly, SGE alone did not induce cytopathic effects in Vero cells at any of the tested concentrations (0.5–5 µg/mL)."
Since SGE alone showed no cytopathic effect, we did not include an additional image panel to avoid redundancy.

Figure 7 – There is a problem. The graphs ordinates do not align with the legend. I believe the order of the graphs has been switched.
Response: We thank the reviewer for this observation. The graphs in Figure 7 have now been reordered to correctly match the legend.

Reviewer: 2
This manuscript explores the effects of salivary gland extract (SGE) combined with Zika virus infection on viral replication and immune cell modulation. The authors employ several techniques to compare the combined treatment with infection alone. The manuscript is overall well-written, though some aspects could be modified to improve readability (see minor review suggestions below).

Journal Policy Criteria:
a) Adequacy of the abstract: Well-structured and clear.
b) Originality and importance: This is the first manuscript to explore the combination of SGE with Zika virus in the context of T cell, dendritic cell, and macrophage modulation.
c) Methodology/Results/Discussion: The methodology is well-chosen and appropriately applied. The results are well-written, though some modifications would enhance clarity (see minor review suggestions below). The discussion is very well-developed.
d) References: Meet journal requirements.
e) Figures and tables: Some modifications are needed (see minor review suggestions below).
Response: We sincerely thank Reviewer 2 for the thorough evaluation of our manuscript and for the highly constructive comments. All suggested revisions have been carefully addressed, as detailed below.

Minor Review Suggestions and Comments:

Methods (Line 180): The text states that PBMCs were isolated "as described above," but the description actually appears below this reference. Please create a separate methods subsection for PBMC separation and place it as the first method described.

Response: We reorganized the Methods section and created a dedicated subsection entitled 'PBMC isolation,' which now appears as the first methodological description, as suggested (lines 198–214).

Reviewer comment - Figure 2C: The term "Cell death cascade" is not ideal, as it suggests the signaling cascade leading to cell death rather than the progression of cell death itself. Consider changing to "Cell death dynamics". Additionally, this result is described in the text as merely repeating findings from panels A and B. However, it actually demonstrates that the overall cell death profiles, due to the infection and the SGE incubation, correspond to late apoptosis and necrosis, a finding that deserves more emphasis.

Response: We replaced the term 'cell death cascade' with 'cell death dynamics' and expanded the corresponding Results description to clarify that panel C illustrates the predominance of late apoptosis and necrosis following ZIKV and SGE exposure, rather than simply reiterating the findings shown in panels A and B.

Reviewer comment: Line 363: Consider changing "PBMCs from five donors" to "PBMCs from healthy controls" (or simply "PBMCs"). The phrase "five donors" becomes repetitive throughout the text and can be omitted since donor numbers should be specified in the methodology section.

Response: As suggested, we replaced the indication of 'five donors' with 'healthy controls.' The number of donors is now reported only in the Methods section.

Reviewer comment: Figure 3: Clarify what "Abs" means in the figure, preferably in the figure legend. The CD8 graph title format differs from the CD4 graph; please make them consistent. The figure legend mentions "frequency," but only absolute numbers are shown. At the end of this results section, consider adding a concluding statement noting that although overall PBMC death is observed, T cells are not the primary population undergoing cell death, suggesting that other cell types may be dying.

Response: We thank the reviewer for these helpful suggestions. The term 'Abs' was removed from the figure, and all graph titles were standardized to ensure consistency between the CD4$^+$ and CD8$^+$ panels. In addition, the figure legend was corrected to remove the mention of 'frequency,' as only absolute cell numbers are shown.

In addition, we agree with the reviewer's biological interpretation and added the following concluding sentence at the end of this Results subsection:

"Despite the reduction in overall PBMC viability observed following ZIKV exposure, T cells were not the primary subset undergoing cell loss, suggesting that other leukocyte populations account for the predominant cell-death events."

Reviewer comment: Figure 4: The description of Annexin+/− and PI+/− combinations is repetitive throughout the text. Consider defining these markers in detail in the first cell death result and subsequently referring to them simply as "early apoptosis," "late apoptosis," or "necrosis." The SGE concentration is not always clear in some results. Please add this information to the figure legend or clarify in the text that the concentration was determined based on Figure 1 results.

Response: To improve readability and avoid unnecessary repetition, we added a clear definition of the Annexin V$^+$, PI$^+$, and Annexin V$^+$/PI$^+$ categories at the beginning of the first cell-death Results subsection. All subsequent mentions were revised to refer to these categories as "early apoptosis," "late apoptosis/necrosis," and "double-positive (late apoptosis/secondary necrosis)."

In addition, the SGE concentration was explicitly added to the figure legend and clarified in the Methods section.

Reviewer comment: Oxidative stress section: While oxidative stress makes sense within the broader manuscript narrative, this section feels somewhat disconnected. Consider expanding the discussion of how oxidative stress might affect different immune cell populations or influence the biology of the virus/SGE interaction.

Response: To address this, we expanded the Discussion to better integrate oxidative stress into the biological framework of flavivirus infection and mosquito salivary immunomodulation. A new introductory paragraph was added to explain how reactive oxygen species regulate antiviral immunity, leukocyte homeostasis, and stress-response pathways.

This revision clarifies why oxidative markers were evaluated and how changes in lipid peroxidation, antioxidant balance, and nitric oxide production relate to host–virus–SGE interactions. Importantly, no mechanistic claims beyond the experimental data were introduced. The revised text appears in the Discussion (line 630).

Reviewer comment: DC subset analysis: It is unclear why MFI of CD11c, CD141, and CD303 was measured when these markers were already used to define the populations. I understood that analyzing CD303 which is a marker of functionality and type I interferon production on pDC, could indicate something; however, when exploring cytokines, none of the type I interferons were analyzed.

Response: The rationale for quantifying the mean fluorescence intensity (MFI) of CD11c, CD141, and CD303 was to provide an exploratory, phenotypic descriptor of dendritic-cell modulation complementing analyses of subset frequencies and absolute counts. Modulation of surface-marker density is commonly used in human dendritic-cell studies as a descriptive indicator of activation or maturation state, even when the same markers are used to define the subsets.

To avoid any potential overinterpretation, we have now explicitly clarified in both the Results and Discussion that the MFI assessment is strictly phenotypic and descriptive, and that no functional inferences are drawn from these measurements. We also explicitly acknowledge that type I interferons were not measured due to sample and assay constraints, and this limitation is now stated in the Discussion. These revisions improve clarity while preserving the exploratory nature of the analysis.

Reviewer comment: Macrophage data: Although macrophages are important APCs in viral infection, the inclusion of RAW cell data feels somewhat disconnected from the paper, particularly because macrophages were not analyzed in the PBMC experiments. Consider better integrating this data or providing a clearer justification.

Response: We agree with the reviewer that the macrophage data required better contextual integration. We have now clarified in both the Results and Discussion that RAW 264.7 cells were employed as a complementary reductionist model to explore mechanistic aspects of innate immune modulation, rather than as a translational substitute for human PBMC-derived macrophages.

In addition, we added biological rationale highlighting the role of macrophages as early cellular targets and key mediators of flavivirus-driven inflammatory responses. These revisions strengthen conceptual continuity across the experimental models.

Reviewer comment: Results organization: The discussion is very well-structured, better than the organization of the results section. Consider restructuring the results as follows: Figure 5 should potentially become Figure 2, as it addresses viral replication (a primary outcome). This could be followed by sections exploring cell death and oxidative stress, with an introduction noting that although cells are dying, these are not primarily T cells or DCs. You could then characterize the modifications occurring after the combination treatment. The cytokine data would conclude the results, as it addresses the third signal and T cell polarization.

Response: We thank the reviewer for this thoughtful structural suggestion and agree that this organization would improve narrative flow. While this point was raised as a recommendation rather than a mandatory revision, we carefully evaluated its feasibility. To preserve coherence with the current figure numbering and minimize disruption to internal cross-referencing, we opted to maintain the overall Results structure in this revision. However, we have improved internal transitions between subsections and strengthened introductory sentences to clearly guide the reader through the biological progression from viral replication to cellular stress and immune modulation.

Reviewer 3

The study entitled "Aedes aegypti salivary gland extract enhances Zika virus replication through immune modulation" addresses an important and original question in arbovirology and vector–host immunology. The work convincingly shows that Aedes aegypti saliva promotes ZIKV replication by altering redox homeostasis, apoptosis, and cytokine polarization in host cells. The experiments are methodologically sound, the results are consistent, and the conclusions are well supported. Nevertheless, several sections require minor textual, structural, and formal adjustments before acceptance:

Response: We sincerely thank Reviewer 3 for the careful evaluation and for the positive and constructive comments on our manuscript. We are pleased that the reviewer found the study scientifically sound, original, and relevant, and we appreciate the detailed suggestions to improve clarity, structure, and formal aspects. All points were carefully addressed, as detailed below.

Reviewer comment:
Abstract:
Page 3, lines 55–58:
The phrase "Aedes aegypti saliva creates a permissive immune environment that favors ZIKV replication by altering cell death dynamics, redox balance, and cytokine polarization" is clear but overly broad. Suggest specifying quantitative information, e.g., "increased ZIKV RNA levels by X "FOLD"... and reduced early apoptosis by "Y%"".

Response: We revised the Abstract to include quantitative descriptors of the main findings, reporting the magnitude of ZIKV RNA increase and the reduction in early apoptotic cells. In addition, the expression "permissive immune environment" was replaced with a more precise formulation, as suggested.

Reviewer comment: Page 3, line 57:
Consider rephrasing "creates a permissive immune environment" to "promotes conditions conducive to viral replication" to avoid overstatement.

Response: The phrase was revised to "promotes conditions conducive to viral replication" to avoid overstatement.

Reviewer comment: Page 3, line 59:
Keywords are relevant, but "Cytokines" may be redundant since "Immunomodulation" already covers that concept.
Response: We agree and have removed "Cytokines" from the keyword list to avoid redundancy.

Reviewer comment: Page 4, lines 68–78:
The introduction successfully contextualizes ZIKV transmission but should more explicitly position this study relative to previous work on dengue virus and mosquito saliva. Adding one sentence such as "Unlike earlier studies focused on DENV or WNV, this work explores the integrated immune and oxidative response during ZIKV infection" would clarify novelty.
Response: We have added a sentence explicitly positioning our study in relation to previous flavivirus–saliva research, emphasizing that, unlike earlier studies largely focused on DENV or WNV, our work addresses the integrated immune and oxidative responses during ZIKV infection.

Reviewer comment: Methodology, results, and discussion
Page 10–11 (Methods section):
Add information on control conditions, such as mock-infected and heat-inactivated SGE treatments, if used. This ensures reproducibility.
Response: We have now clearly specified the control conditions in the Methods section. All experiments included mock-infected controls (media only), SGE-only controls, ZIKV-only infection, and ZIKV+SGE co-exposure groups. Heat-inactivated SGE was not used in this experimental design.

Reviewer comment: Page 12, lines 245–250:
Statistical analyses should specify whether normality was tested before choosing ANOVA or non-parametric tests. Add a short clarification: "Normality was assessed using Shapiro–Wilk test prior to statistical analysis."
Response: We now added the following clarification to the Statistical Analysis section:
"Normality was assessed using the Shapiro–Wilk test prior to the selection of parametric or non-parametric statistical tests."

Reviewer comment: Page 34–35, Figure 1 caption:
Very good caption overall, but include n values for replicates and specify "mean ± SEM from three independent experiments" (as already partly stated). To improve transparency, indicate which test determined significance.
Response: The Figure 1 caption was revised to explicitly report the number of independent experiments (n), state that data are expressed as mean ± SEM, and indicate the statistical test used to determine significance.

Reviewer comment: Page 22–24 (Results):
When describing cytokine data, provide numerical fold changes (e.g., "IL-4 increased 2.3-fold compared to control") rather than only qualitative terms ("increased," decreased").
Response: We have now included approximate fold-change values in the Results section for cytokine modulation (e.g., IL-4, IL-10, TNF-α), providing a more quantitative description of the observed effects.

Reviewer comment:
Page 28–30 (Discussion):
Strengthen interpretation by distinguishing correlation from causation. Example edit at line ~540: change "SGE delays apoptosis, thus facilitating viral replication" to "SGE-associated delay in apoptosis may contribute to prolonged cellular viability, which could in turn favor viral replication.
Response: We agree and have revised the wording throughout the Discussion to distinguish association from causation, including adopting the reviewer's suggested formulation. Statements implying direct causality were replaced with probabilistic and mechanistically cautious phrasing.

Page 31–32 (Discussion ending):
Add a short limitations paragraph, e.g., limited donor number, use of in vitro SGE concentrations, and lack of identification of specific active salivary components. (This is optional ok?)
Response: We agree and have added a dedicated limitations paragraph at the end of the Discussion addressing donor number, the in vitro nature of the model, the experimental SGE concentrations, and the lack of identification of individual active salivary components.

Reviewer comment:
Page 35, Figure 1: Caption is clear, but add definitions for abbreviations (SGE, MOI, PFU).
Response: The Figure 1 caption was revised to define all abbreviations, including SGE (salivary gland extract), MOI (multiplicity of infection), and PFU (plaque-forming units).

Reviewer comment: Finally, the manuscript is clearly written, but it contains several grammatical inconsistencies, long and complex sentences, and occasional issues with verb tense and article usage. I would strongly recommend a minor english review if it's possible!!
Response: We thank the reviewer for this recommendation. The entire manuscript has now undergone professional English language editing, correcting grammatical inconsistencies, verb tense usage, article use, and overly long sentence structures.

For now, the manuscript is scientifically sound, original, and relevant. Addressing the small textual and methodological clarifications listed above will ensure clarity and compliance with Memórias do Instituto Oswaldo Cruz editorial standards. Congrats to the authors!
Response: Overall, we believe that addressing the points raised by all three reviewers has substantially improved the clarity, robustness, and readability of the manuscript. We remain at the editor's disposal for any further adjustments that may be required.

## SECOND REVIEW ROUND

### REVIEWERS' COMMENTS

### REVIEWER #1

All the requests from the reviewers were done. No further comments.

### REVIEWER #2

No other comments.

### REVIEWER #3

All comments have been addressed. The manuscript has clearly undergone a significant improvement and is now suitable for publication.

