## [Reviewer Report · FIRST REVIEW ROUND - REVIEWERS COMMENTS]

## REVIEWER #1

The authors conducted an evaluation of the effects of Aedes aegypti salivary glands extracts (SGE) on Zika virus infection.

Overall the manuscript is well written and the assays support most of the conclusions.

Below there are a few comments that, hopefully, will be useful for the authors.

General comments

The authors found that SGE presence resulted in delayed peripheral blood mononuclear cells (PBMCs) apoptosis.

They also have found that SGE enhances Zikv replication in PBMCs (Line 384) and stated that SGE modulation of cell machinery for replication (Lines 405 and 544) may be a possible explanation of higher viral load.

However the higher viral load could not be a result of higher PBMC cell survivability (and, therefore, greater cell count) rather than cell machinery modulation?

If that is the case this possibility should be addressed as well in the replication assay.

Specific comments

Line 156 - Plaque

Line 170 - This was a bit confusing. I believe the viral infection and SGE treatment were simultaneous in this line and in other assays as well.

But the way as it is written may also imply a prior treatment with SGE with subsequent viral infection.

I would recommend a clarification on this line and in others as well

Line 243 - There are two final points

Line 311 - Plaque

Line 308 - Do SGE alone result in cytopathic effect?

If so it would be good to have an image for comparison in figure 1

Figure 7 - There is a problem. The graphs ordinates do not align with the legend.

I believe the order of the graphs has been switched.

## REVIEWER #2

This manuscript explores the effects of salivary gland extract (SGE) combined with Zika virus infection on viral replication and immune cell modulation.

The authors employ several techniques to compare the combined treatment with infection alone.

The manuscript is overall well-written, though some aspects could be modified to improve readability (see minor review suggestions below).

Journal Policy Criteria:

a) Adequacy of the abstract: Well-structured and clear.

b) Originality and importance: This is the first manuscript to explore the combination of SGE with Zika virus in the context of T cell, dendritic cell, and macrophage modulation.

c) Methodology/Results/Discussion: The methodology is well-chosen and appropriately applied. The results are well-written, though some modifications would enhance clarity (see minor review suggestions below).

The discussion is very well-developed.

d) References: Meet journal requirements.

e) Figures and tables: Some modifications are needed (see minor review suggestions below).

Minor Review Suggestions and Comments:

Methods (Line 180): The text states that PBMCs were isolated “as described above,” but the description actually appears below this reference.

Please create a separate methods subsection for PBMC separation and place it as the first method described.

Figure 2C: The term “Cell death cascade” is not ideal, as it suggests the signaling cascade leading to cell death rather than the progression of cell death itself.

Consider changing to “Cell death dynamics”. Additionally, this result is described in the text as merely repeating findings from panels A and B. However, it actually demonstrates that the overall cell death profiles, due to the infection and the SGE incubation, correspond to late apoptosis and necrosis, a finding that deserves more emphasis.

Line 363: Consider changing “PBMCs from five donors” to “PBMCs from healthy controls” (or simply “PBMCs”).

The phrase “five donors” becomes repetitive throughout the text and can be omitted since donor numbers should be specified in the methodology section.

Figure 3:

Clarify what “Abs” means in the figure, preferably in the figure legend.

The CD8 graph title format differs from the CD4 graph; please make them consistent.

The figure legend mentions “frequency,” but only absolute numbers are shown.

At the end of this results section, consider adding a concluding statement noting that although overall PBMC death is observed, T cells are not the primary population undergoing cell death, suggesting that other cell types may be dying.

Figure 4:

The description of Annexin+/− and PI+/− combinations is repetitive throughout the text.

Consider defining these markers in detail in the first cell death result and subsequently referring to them simply as “early apoptosis,” “late apoptosis,” or “necrosis.”

The SGE concentration is not always clear in some results.

Please add this information to the figure legend or clarify in the text that the concentration was determined based on Figure 1 results.

Oxidative stress section: While oxidative stress makes sense within the broader manuscript narrative, this section feels somewhat disconnected.

Consider expanding the discussion of how oxidative stress might affect different immune cell populations or influence the biology of the virus/SGE interaction.

DC subset analysis: It is unclear why MFI of CD11c, CD141, and CD303 was measured when these markers were already used to define the populations.

I understood that analyzing CD303 which is a marker of functionality and type I interferon production on pDC, could indicate something; however, when exploring cytokines, none of the type I interferons were analyzed.

Macrophage data: Although macrophages are important APCs in viral infection, the inclusion of RAW cell data feels somewhat disconnected from the paper, particularly because macrophages were not analyzed in the PBMC experiments.

Consider better integrating this data or providing a clearer justification.

Results organization: The discussion is very well-structured, better than the organization of the results section.

Consider restructuring the results as follows: Figure 5 should potentially become Figure 2, as it addresses viral replication (a primary outcome).

This could be followed by sections exploring cell death and oxidative stress, with an introduction noting that although cells are dying, these are not primarily T cells or DCs.

You could then characterize the modifications occurring after the combination treatment.

The cytokine data would conclude the results, as it addresses the third signal and T cell polarization.

Summary of my comments:

Comments 1–7 are necessary revisions.

Comment 8 (results reorganization) is a suggestion that would significantly improve the manuscript’s flow but is not essential.

## REVIEWER #3

The study entitled “Aedes aegypti salivary gland extract enhances Zika virus replication through immune modulation” addresses an important and original question in arbovirology and vector–host immunology.

The work convincingly shows that Aedes aegypti saliva promotes ZIKV replication by altering redox homeostasis, apoptosis, and cytokine polarization in host cells.

The experiments are methodologically sound, the results are consistent, and the conclusions are well supported.

Nevertheless, several sections require minor textual, structural, and formal adjustments before acceptance:

Abstract:

Page 3, lines 55–58:

The phrase “Aedes aegypti saliva creates a permissive immune environment that favors ZIKV replication by altering cell death dynamics, redox balance, and cytokine polarization” is clear but overly broad.

Suggest specifying quantitative information, e.g., “increased ZIKV RNA levels by X “FOLD”... and reduced early apoptosis by “Y%””.

Page 3, line 57:

Consider rephrasing “creates a permissive immune environment” to “promotes conditions conducive to viral replication” to avoid overstatement.

Page 3, line 59:

Keywords are relevant, but “Cytokines” may be redundant since “Immunomodulation” already covers that concept.

Page 4, lines 68–78:

The introduction successfully contextualizes ZIKV transmission but should more explicitly position this study relative to previous work on dengue virus and mosquito saliva.

Adding one sentence such as “Unlike earlier studies focused on DENV or WNV, this work explores the integrated immune and oxidative response during ZIKV infection” would clarify novelty.

Methodology, results, and discussion

Page 10–11 (Methods section):

Add information on control conditions, such as mock-infected and heat-inactivated SGE treatments, if used. This ensures reproducibility.

Page 12, lines 245–250:

Statistical analyses should specify whether normality was tested before choosing ANOVA or non-parametric tests.

Add a short clarification: “Normality was assessed using Shapiro–Wilk test prior to statistical analysis.”

Page 34–35, Figure 1 caption:

Very good caption overall, but include n values for replicates and specify “mean ± SEM from three independent experiments” (as already partly stated).

To improve transparency, indicate which test determined significance.

Page 22–24 (Results):

When describing cytokine data, provide numerical fold changes (e.g., “IL-4 increased 2.3-fold compared to control”) rather than only qualitative terms (“increased,” “decreased”).

Page 28–30 (Discussion):

Strengthen interpretation by distinguishing correlation from causation. Example edit at line ~540: change “SGE delays apoptosis, thus facilitating viral replication” to “SGE-associated delay in apoptosis may contribute to prolonged cellular viability, which could in turn favor viral replication.”

Page 31–32 (Discussion ending):

Add a short limitations paragraph, e.g., limited donor number, use of in vitro SGE concentrations, and lack of identification of specific active salivary components.

(This is optional ok?)

Regarding the figures I have a minor suggestion:

Page 35, Figure 1:

Caption is clear, but add definitions for abbreviations (SGE, MOI, PFU).

Finally, the manuscript is clearly written, but it contains several grammatical inconsistencies, long and complex sentences, and occasional issues with verb tense and article usage.

I would strongly recommend a minor english review if it’s possible!!

For now, the manuscript is scientifically sound, original, and relevant.

Addressing the small textual and methodological clarifications listed above will ensure clarity and compliance with Memórias do Instituto Oswaldo Cruz editorial standards.

Congrats to the authors!

## AUTHORS’ RESPONSE TO THE REVIEWERS

Dear Editor

Thank you for handling the revision of the manuscript “Aedes aegypti salivary gland extract enhances Zika virus replication through immune modulation.”

In response to the reviewers’ comments, we have revised the manuscript accordingly.

We believe that all suggestions have been fully addressed. Below, we provide a detailed, point-by-point response to each reviewer.

All line numbers mentioned refer to the revised version of the manuscript.

Reviewer: 1

The authors conducted an evaluation of the effects of Aedes aegypti salivary glands extracts (SGE) on Zika virus infection.

Overall the manuscript is well written and the assays support most of the conclusions.

Below there are a few comments that, hopefully, will be useful for the authors.

Dear Reviewer, #1,

We thank you for the time devoted to evaluating our manuscript.

Your comments have helped improve the quality of our work. Below, we provide our responses to each of your observations.

General comments

The authors found that SGE presence resulted in delayed peripheral blood mononuclear cells (PBMCs) apoptosis.

They also have found that SGE enhances Zikv replication in PBMCs (Line 384) and stated that SGE modulation of cell machinery for replication (Lines 405 and 544) may be a possible explanation of higher viral load.

However the higher viral load could not be a result of higher PBMC cell survivability (and, therefore, greater cell count) rather than cell machinery modulation?

If that is the case this possibility should be addressed as well in the replication assay.

Response: We thank the reviewer for raising this important point.

We clarify that all PBMC cultures were seeded with the same initial number of cells (1 × 10⁶ cells/well), ensuring that baseline cell numbers were rigorously controlled across all experimental conditions.

We agree that the observed increase in viral load may partially result from enhanced PBMC survival.

The increased survival of PBMCs and dendritic cells in the presence of SGE, as supported by our Annexin V/PI data, could indeed contribute to a higher overall viral output by preserving a larger pool of susceptible target cells.

It is also possible that both mechanisms operate simultaneously.

Importantly, our findings are more consistent with a combined mechanism in which SGE not only preserves cell viability but also modulates antiviral pathways, rather than solely increasing intrinsic viral replication on a per-cell basis.

We have addressed this possibility in the Discussion section (line 653 of the revised version).

“It is also important to consider that the increased viral RNA detected in ZIKV+SGE cultures may partially reflect the preservation of susceptible leukocytes rather than an isolated increase in intrinsic replication on a per-cell basis. In our experimental design, all PBMC conditions were initiated with identical cell numbers; however, SGE attenuated apoptotic and necrotic death in specific subsets, particularly dendritic cells and CD4⁺ T cells, which could extend the time window during which these cells remain permissive to infection. Therefore, the higher viral yields observed in the presence of SGE are likely the result of a combined effect of target-cell preservation and active modulation of antiviral pathways, as suggested by previous reports showing that mosquito salivary components alter host cell death, inflammasome activation, and interferon responses during flavivirus infection”(Page 27, lines 653 to 662).

Specific Comments

Line 156 – “Plaque”

Response: The word “plaque” has been corrected.

Line 170 – This was a bit confusing. I believe the viral infection and SGE treatment were simultaneous in this line and in other assays as well.

But the way as it is written may also imply a prior treatment with SGE with subsequent viral infection.

I would recommend a clarification on this line and in others as well

Response: In all assays, ZIKV and SGE were added simultaneously, and cells were not pretreated with SGE.

This point has been clarified in the Methods section. The revised text (line 189) now reads:

“ZIKV (1×10³ PFU/mL) and SGE (0.5–5 µg/mL) were added simultaneously to the cultures.”

To avoid any ambiguity throughout the manuscript, we have also added the clarification “ZIKV and SGE were added to the cultures at the same time” at lines 292, 306 and 318.

Line 243 – There are two final points

Response: The sentence was edited for clarity, as suggested.

Line 311 – “Plaque”

Response: The word “plaque” has been corrected.

Line 308 – Do SGE alone result in cytopathic effect?

If so it would be good to have an image for comparison in figure 1

Response: In our assays, SGE alone did not induce visible cytopathic effects in Vero cells at any of the tested concentrations (0.5–5 µg/mL).

This information has now been clarified in the Results (line 358):

“Importantly, SGE alone did not induce cytopathic effects in Vero cells at any of the tested concentrations (0.5–5 µg/mL).”

Since SGE alone showed no cytopathic effect, we did not include an additional image panel to avoid redundancy.

Figure 7 – There is a problem. The graphs ordinates do not align with the legend.

I believe the order of the graphs has been switched.

Response: We thank the reviewer for this observation. The graphs in Figure 7 have now been reordered to correctly match the legend.

Reviewer: 2

This manuscript explores the effects of salivary gland extract (SGE) combined with Zika virus infection on viral replication and immune cell modulation.

The authors employ several techniques to compare the combined treatment with infection alone.

The manuscript is overall well-written, though some aspects could be modified to improve readability (see minor review suggestions below).

Journal Policy Criteria:

a) Adequacy of the abstract: Well-structured and clear.

b) Originality and importance: This is the first manuscript to explore the combination of SGE with Zika virus in the context of T cell, dendritic cell, and macrophage modulation.

c) Methodology/Results/Discussion: The methodology is well-chosen and appropriately applied. The results are well-written, though some modifications would enhance clarity (see minor review suggestions below).

The discussion is very well-developed.

d) References: Meet journal requirements.

e) Figures and tables: Some modifications are needed (see minor review suggestions below).

Response: We sincerely thank Reviewer 2 for the thorough evaluation of our manuscript and for the highly constructive comments.

All suggested revisions have been carefully addressed, as detailed below.

Minor Review Suggestions and Comments:

Methods (Line 180): The text states that PBMCs were isolated “as described above,” but the description actually appears below this reference.

Please create a separate methods subsection for PBMC separation and place it as the first method described.

Response: We reorganized the Methods section and created a dedicated subsection entitled ‘PBMC isolation,’ which now appears as the first methodological description, as suggested (lines 198–214).

Reviewer comment - Figure 2C: The term “Cell death cascade” is not ideal, as it suggests the signaling cascade leading to cell death rather than the progression of cell death itself.

Consider changing to “Cell death dynamics”. Additionally, this result is described in the text as merely repeating findings from panels A and B. However, it actually demonstrates that the overall cell death profiles, due to the infection and the SGE incubation, correspond to late apoptosis and necrosis, a finding that deserves more emphasis.

Response: We replaced the term ‘cell death cascade’ with ‘cell death dynamics’ and expanded the corresponding Results description to clarify that panel C illustrates the predominance of late apoptosis and necrosis following ZIKV and SGE exposure, rather than simply reiterating the findings shown in panels A and B.

Reviewer comment: Line 363: Consider changing “PBMCs from five donors” to “PBMCs from healthy controls” (or simply “PBMCs”).

The phrase “five donors” becomes repetitive throughout the text and can be omitted since donor numbers should be specified in the methodology section.

Response: As suggested, we replaced the indication of ‘five donors’ with ‘healthy controls.’ The number of donors is now reported only in the Methods section.

Reviewer comment: Figure 3: Clarify what “Abs” means in the figure, preferably in the figure legend.

The CD8 graph title format differs from the CD4 graph; please make them consistent.

The figure legend mentions “frequency,” but only absolute numbers are shown.

At the end of this results section, consider adding a concluding statement noting that although overall PBMC death is observed, T cells are not the primary population undergoing cell death, suggesting that other cell types may be dying.

Response: We thank the reviewer for these helpful suggestions. The term ‘Abs’ was removed from the figure, and all graph titles were standardized to ensure consistency between the CD4⁺ and CD8⁺ panels.

In addition, the figure legend was corrected to remove the mention of ‘frequency,’ as only absolute cell numbers are shown.

In addition, we agree with the reviewer’s biological interpretation and added the following concluding sentence at the end of this Results subsection:

“Despite the reduction in overall PBMC viability observed following ZIKV exposure, T cells were not the primary subset undergoing cell loss, suggesting that other leukocyte populations account for the predominant cell-death events.”

Reviewer comment: Figure 4: The description of Annexin+/− and PI+/− combinations is repetitive throughout the text.

Consider defining these markers in detail in the first cell death result and subsequently referring to them simply as “early apoptosis,” “late apoptosis,” or “necrosis.”

The SGE concentration is not always clear in some results.

Please add this information to the figure legend or clarify in the text that the concentration was determined based on Figure 1 results.

Response: To improve readability and avoid unnecessary repetition, we added a clear definition of the Annexin V⁺, PI⁺, and Annexin V⁺/PI⁺ categories at the beginning of the first cell-death Results subsection.

All subsequent mentions were revised to refer to these categories as “early apoptosis,” “late apoptosis/necrosis,” and “double-positive (late apoptosis/secondary necrosis).”

In addition, the SGE concentration was explicitly added to the figure legend and clarified in the Methods section.

Reviewer comment: Oxidative stress section: While oxidative stress makes sense within the broader manuscript narrative, this section feels somewhat disconnected.

Consider expanding the discussion of how oxidative stress might affect different immune cell populations or influence the biology of the virus/SGE interaction.

Response: To address this, we expanded the Discussion to better integrate oxidative stress into the biological framework of flavivirus infection and mosquito salivary immunomodulation.

A new introductory paragraph was added to explain how reactive oxygen species regulate antiviral immunity, leukocyte homeostasis, and stress-response pathways.

This revision clarifies why oxidative markers were evaluated and how changes in lipid peroxidation, antioxidant balance, and nitric oxide production relate to host–virus–SGE interactions.

Importantly, no mechanistic claims beyond the experimental data were introduced. The revised text appears in the Discussion (line 630).

Reviewer comment: DC subset analysis: It is unclear why MFI of CD11c, CD141, and CD303 was measured when these markers were already used to define the populations.

I understood that analyzing CD303 which is a marker of functionality and type I interferon production on pDC, could indicate something; however, when exploring cytokines, none of the type I interferons were analyzed.

Response: The rationale for quantifying the mean fluorescence intensity (MFI) of CD11c, CD141, and CD303 was to provide an exploratory, phenotypic descriptor of dendritic-cell modulation complementing analyses of subset frequencies and absolute counts.

Modulation of surface-marker density is commonly used in human dendritic-cell studies as a descriptive indicator of activation or maturation state, even when the same markers are used to define the subsets.

To avoid any potential overinterpretation, we have now explicitly clarified in both the Results and Discussion that the MFI assessment is strictly phenotypic and descriptive, and that no functional inferences are drawn from these measurements.

We also explicitly acknowledge that type I interferons were not measured due to sample and assay constraints, and this limitation is now stated in the Discussion.

These revisions improve clarity while preserving the exploratory nature of the analysis.

Reviewer comment: Macrophage data: Although macrophages are important APCs in viral infection, the inclusion of RAW cell data feels somewhat disconnected from the paper, particularly because macrophages were not analyzed in the PBMC experiments.

Consider better integrating this data or providing a clearer justification.

Response: We agree with the reviewer that the macrophage data required better contextual integration.

We have now clarified in both the Results and Discussion that RAW 264.7 cells were employed as a complementary reductionist model to explore mechanistic aspects of innate immune modulation, rather than as a translational substitute for human PBMC-derived macrophages.

In addition, we added biological rationale highlighting the role of macrophages as early cellular targets and key mediators of flavivirus-driven inflammatory responses.

These revisions strengthen conceptual continuity across the experimental models.

Reviewer comment: Results organization: The discussion is very well-structured, better than the organization of the results section.

Consider restructuring the results as follows: Figure 5 should potentially become Figure 2, as it addresses viral replication (a primary outcome).

This could be followed by sections exploring cell death and oxidative stress, with an introduction noting that although cells are dying, these are not primarily T cells or DCs.

You could then characterize the modifications occurring after the combination treatment.

The cytokine data would conclude the results, as it addresses the third signal and T cell polarization.

Response: We thank the reviewer for this thoughtful structural suggestion and agree that this organization would improve narrative flow.

While this point was raised as a recommendation rather than a mandatory revision, we carefully evaluated its feasibility.

To preserve coherence with the current figure numbering and minimize disruption to internal cross-referencing, we opted to maintain the overall Results structure in this revision.

However, we have improved internal transitions between subsections and strengthened introductory sentences to clearly guide the reader through the biological progression from viral replication to cellular stress and immune modulation.

Reviewer 3

The study entitled “Aedes aegypti salivary gland extract enhances Zika virus replication through immune modulation” addresses an important and original question in arbovirology and vector–host immunology.

The work convincingly shows that Aedes aegypti saliva promotes ZIKV replication by altering redox homeostasis, apoptosis, and cytokine polarization in host cells.

The experiments are methodologically sound, the results are consistent, and the conclusions are well supported.

Nevertheless, several sections require minor textual, structural, and formal adjustments before acceptance:

Response: We sincerely thank Reviewer 3 for the careful evaluation and for the positive and constructive comments on our manuscript.

We are pleased that the reviewer found the study scientifically sound, original, and relevant, and we appreciate the detailed suggestions to improve clarity, structure, and formal aspects.

All points were carefully addressed, as detailed below.

Reviewer comment:

Abstract:

Page 3, lines 55–58:

The phrase “Aedes aegypti saliva creates a permissive immune environment that favors ZIKV replication by altering cell death dynamics, redox balance, and cytokine polarization” is clear but overly broad.

Suggest specifying quantitative information, e.g., “increased ZIKV RNA levels by X “FOLD”... and reduced early apoptosis by “Y%””.

Response: We revised the Abstract to include quantitative descriptors of the main findings, reporting the magnitude of ZIKV RNA increase and the reduction in early apoptotic cells.

In addition, the expression “permissive immune environment” was replaced with a more precise formulation, as suggested.

Reviewer comment: Page 3, line 57:

Consider rephrasing “creates a permissive immune environment” to “promotes conditions conducive to viral replication” to avoid overstatement.

Response: The phrase was revised to “promotes conditions conducive to viral replication” to avoid overstatement.

Reviewer comment: Page 3, line 59:

Keywords are relevant, but “Cytokines” may be redundant since “Immunomodulation” already covers that concept.

Response: We agree and have removed “Cytokines” from the keyword list to avoid redundancy.

Reviewer comment: Page 4, lines 68–78:

The introduction successfully contextualizes ZIKV transmission but should more explicitly position this study relative to previous work on dengue virus and mosquito saliva.

Adding one sentence such as “Unlike earlier studies focused on DENV or WNV, this work explores the integrated immune and oxidative response during ZIKV infection” would clarify novelty.

Response: We have added a sentence explicitly positioning our study in relation to previous flavivirus–saliva research, emphasizing that, unlike earlier studies largely focused on DENV or WNV, our work addresses the integrated immune and oxidative responses during ZIKV infection.

Reviewer comment: Methodology, results, and discussion

Page 10–11 (Methods section):

Add information on control conditions, such as mock-infected and heat-inactivated SGE treatments, if used. This ensures reproducibility.

Response: We have now clearly specified the control conditions in the Methods section.

All experiments included mock-infected controls (media only), SGE-only controls, ZIKV-only infection, and ZIKV+SGE co-exposure groups.

Heat-inactivated SGE was not used in this experimental design.

Reviewer comment: Page 12, lines 245–250:

Statistical analyses should specify whether normality was tested before choosing ANOVA or non-parametric tests.

Add a short clarification: “Normality was assessed using Shapiro–Wilk test prior to statistical analysis.”

Response: We now added the following clarification to the Statistical Analysis section:

“Normality was assessed using the Shapiro–Wilk test prior to the selection of parametric or non-parametric statistical tests.”

Reviewer comment: Page 34–35, Figure 1 caption:

Very good caption overall, but include n values for replicates and specify “mean ± SEM from three independent experiments” (as already partly stated).

To improve transparency, indicate which test determined significance.

Response: The Figure 1 caption was revised to explicitly report the number of independent experiments (n), state that data are expressed as mean ± SEM, and indicate the statistical test used to determine significance.

Reviewer comment: Page 22–24 (Results):

When describing cytokine data, provide numerical fold changes (e.g., “IL-4 increased 2.3-fold compared to control”) rather than only qualitative terms (“increased,” decreased”).

Response: We have now included approximate fold-change values in the Results section for cytokine modulation (e.g., IL-4, IL-10, TNF-α), providing a more quantitative description of the observed effects.

Reviewer comment:

Page 28–30 (Discussion):

Strengthen interpretation by distinguishing correlation from causation. Example edit at line ~540: change “SGE delays apoptosis, thus facilitating viral replication” to “SGE-associated delay in apoptosis may contribute to prolonged cellular viability, which could in turn favor viral replication.

Response: We agree and have revised the wording throughout the Discussion to distinguish association from causation, including adopting the reviewer’s suggested formulation.

Statements implying direct causality were replaced with probabilistic and mechanistically cautious phrasing.

Page 31–32 (Discussion ending):

Add a short limitations paragraph, e.g., limited donor number, use of in vitro SGE concentrations, and lack of identification of specific active salivary components.

(This is optional ok?)

Response: We agree and have added a dedicated limitations paragraph at the end of the Discussion addressing donor number, the in vitro nature of the model, the experimental SGE concentrations, and the lack of identification of individual active salivary components.

Reviewer comment:

Page 35, Figure 1: Caption is clear, but add definitions for abbreviations (SGE, MOI, PFU).

Response: The Figure 1 caption was revised to define all abbreviations, including SGE (salivary gland extract), MOI (multiplicity of infection), and PFU (plaque-forming units).

Reviewer comment: Finally, the manuscript is clearly written, but it contains several grammatical inconsistencies, long and complex sentences, and occasional issues with verb tense and article usage.

I would strongly recommend a minor english review if it’s possible!!

Response: We thank the reviewer for this recommendation. The entire manuscript has now undergone professional English language editing, correcting grammatical inconsistencies, verb tense usage, article use, and overly long sentence structures.

For now, the manuscript is scientifically sound, original, and relevant.

Addressing the small textual and methodological clarifications listed above will ensure clarity and compliance with Memórias do Instituto Oswaldo Cruz editorial standards.

Congrats to the authors!

Response: Overall, we believe that addressing the points raised by all three reviewers has substantially improved the clarity, robustness, and readability of the manuscript.

We remain at the editor’s disposal for any further adjustments that may be required.

---

## [Reviewer Report · REVIEWERS COMMENTS]

## REVIEWER #1

All the requests from the reviewers were done. No further comments.

## REVIEWER #2

No other comments.

## REVIEWER #3

All comments have been addressed. The manuscript has clearly undergone a significant improvement and is now suitable for publication.